# Rock glaciers in the Daxue Shan, southeastern Tibetan Plateau: an inventory, their distribution, and their environmental controls

Zeze Ran* and Gengnian Liu

Key Laboratory for Earth Surface Processes of the Ministry of Education, College of Urban and Environmental Sciences, Peking University, Beijing, 100871 China

*Correspondence to: Zeze Ran (ranzeze@pku.edu.cn)

**Abstract.** Rock glaciers are typical periglacial landforms. They can indicate the existence of permafrost, and can also shed light on the regional geomorphological and climatic conditions under which they may have developed. This article provides the first rock glacier inventory of the Daxue Shan. The inventory has been based on analyses of Google Earth imagery. In total, 295 rock glaciers were identified in the Daxue Shan, covering a total area of 55.70 $km^2$, between the altitudes of 4,300 and 4,600 m above sea level (asl). Supported by the ArcGIS and SPSS software programs, we extracted and calculated the parameters of these rock glaciers, and analyzed the characteristics of their spatial distribution within the Daxue Shan. Our inventory suggests that the lower altitudinal boundary for permafrost across the eight aspects of slopes observed in the Daxue Shan differs significantly. The lower altitudinal permafrost boundary is ~104 m lower on eastern- rather than western-facing slopes. The analysis of rock glaciers parameters indicates that the formation of rock glaciers is closely related to local topographical parameters. These results show that environmental controls (*i.e.*, topographical, climatic, lithological factors) greatly affect the formation and development of rock glaciers. This study provides important data for exploring the relation between maritime periglacial environments and the development of rock glaciers on the southeastern Tibetan Plateau (TP). It may also highlight the characteristics typical of rock glaciers found in a maritime setting.

**Keywords:** rock glaciers; inventory; distribution; environmental controls; Daxue Shan

## 1 Introduction

The term 'rock glacier' was first proposed by the American scholar Capps when the investigating Kennicott Glacier in Alaska (Capps, 1910). By definition, rock glaciers consist of perennially frozen masses of ice and debris that creep downslope under the weight of gravity (Haeberli, 1985; Barsch, 1996; Haeberli et al., 2006). As the bodies of rock glaciers are similar to moraines in that, as their ice mass moves over a pore ice surface, they do not sort materials in relation to the thickness of the debris they contain. Many Himalayan rock glaciers develop out of moraines and it is hard to distinguish where the moraine ends and the rock glacier begins. Statistically, rock glaciers occupy extensive areas above the forest line in the mountainous regions of the world (Haeberli, 1985). Indeed, there are ~73,000 rock glaciers in the world (Jones et al., 2018a), with ~1,000 active rock glaciers in the Swiss Alps alone. The ways in which rock glaciers move can significantly influence any engineering and transportation infrastructure in regions affected by permafrost. The freeze-thaw process experienced by the ice masses within rock glaciers can exert a major impact on the hydrological cycle (Azócar and Brenning, 2010; Jones et al., 2018a; Jones et al., 2018b). Rock glacier research may therefore aid a more detailed and accurate understanding of the genesis of periglacial geomorphology and of the ongoing and developmental relation between rock glaciers and their local environments.

Over the last twenty years, with the rapid development of more advanced Geographical Information System (GIS), remote sensing (RS) and statistical techniques, rock glacier research has entered a new, accelerated phase. This phase has included the compilation of rock glacier inventories (*e.g.*, Bolch and Marchenko, 2009; Cremonese et al., 2011; Bolch and Gorbunov, 2014; Falaschi et al., 2014; Colucci et al., 2016; Janke et al., 2017; Wang et al., 2017; Jones et al., 2018a), the mapping of their spatial distributions and their relations with environmental controls such as topography and climate (*e.g.*, Chueca, 1992; Brazier et al., 1998; Brenning, 2005; Janke, 2007; Johnson et al., 2007; Kenner and Magnusson, 2017; Onaca et al., 2017; Jones et al., 2018b), estimations of the distribution of permafrost based on rock glaciers (*e.g.*, Allen et al., 2008; Boeckli et al., 2012; Schmid et al., 2015; Sattler et al., 2016), and the dynamic movement of rock glaciers (*e.g.*, Haeberli et al., 2006; Liu et al., 2013; Muller et al., 2016; Wang et al., 2017). However, compared with ice glaciers, rock glaciers remain poorly described and infrequently studied because they are mixtures of rock fragments of different sizes, and therefore cannot easily be automatically mapped from RS data because they are spectrally similar to their surroundings (Brenning, 2009). Both

supraglacial-debris (upon the glacier) and debris along the glacier margins originate from surrounding valley rock (Jones et al., 2018b), and their debris surface does not produce a distinct spectral signal. As a result, it is often difficult to distinguish relict rock glaciers from inactive rock glaciers that still contain ice using RS imagery (Millar and Westfall, 2008; Kenner and Magnusson, 2017).

Rock glacier research in China has, up to this point, focused principally on the Tianshan Mountains (Cui and Zhu, 1989; Qiu, 1993; Zhu et al., 1996; Wang et al., 2017). However, the study of the rock glaciers of the Daxue Shan on the southeastern margins of the TP is less involved. As this region is located in the transition zone between the TP and the Yangtze Platform, it has been, and continues to be, strongly uplifted and deformed due to the extrusion and collision of the Indian and Eurasian continental plates since the Quaternary. This region is therefore characterized by an extremely complex matrix of relations

between different environmental factors such as climate and geomorphology. It is therefore of particular importance to study the environmental controls on the rock glaciers of the Daxue Shan as an aid to the further study of the complex geographical environment, natural hazards, environmental planning and management found on the southeastern margins of the TP. The purpose of this study was twofold: first, to describe and map the previously undocumented rock glaciers in the Daxue Shan; and second, to complete a systematic inventory of the characteristics and distribution of, and environmental controls on, the

rock glaciers of the Daxue Shan. In addition, there was an analysis and discussion of the mechanisms driving the formation, development and spatial distribution of the rock glaciers of the Daxue Shan rock glaciers in relation to different environmental controls (*i.e.*, climatic, topographical and lithological factors).

## 2 Study Area

The study area is situated in China's Sichuan Province between 29.956°N~30.573°N and 101.477°E~101.974°E (Fig. 1). The

20 Daxue Shan include the Mt. Zheduo (4962 m asl), which lies between the Yarlung and Dadu rivers. To the west is the uplifted eastern sector of the TP, and to the east are mountain gorges, both of which are important geographical boundaries (Zhang et al., 2017). The topography of the Daxue Shan is characterized by the strong downcutting of high energy water courses such as those of the Minjiang and Dadu rivers, resulting in a great altitudinal range (1349 m asl ~ 7321 m asl). The region's climate is

relatively warm and humid, and is strongly influenced by a southwesterly monsoonal atmospheric circulation. East of the Daxue Shan is a subtropical monsoon climatic zone which is principally affected by the aforementioned southwesterly monsoonal atmospheric circulation, but also by a southeasterly monsoonal atmospheric circulation and the Westerlies, all of which transport abundant precipitation to this region. West of the Daxue Shan the subtropical monsoon and continental plateau

climatic zones intersect, producing a cold-temperate climate, as well as abundant precipitation. Geologically, the Daxue Shan are located on the eastern margins of the TP, where the Songpan, Chuandian and South China tectonic blocks intersect. The Xianshuihe (Ganzi-Yushu) Fault passes to the northwest of the Daxue Shan (Zhang, 2013).

## 3 Methods

### 3.1 Rock glacier inventory, classification and database

The availability of more powerful RS tools such as Google Earth has transformed geomorphological fieldwork and has, on the whole, made the recognition of landforms in remote and poorly accessible areas both fast and easy (Slaymaker, 2001; Bolch, 2004; Kaab et al., 2005). An inventory of the rock glaciers of the Daxue Shan was compiled using high-resolution Google Earth satellite imagery (for the period October 2014~January 2017). Google Earth has been previously used for rock glacier identification in the Bolivian Andes (Rangecroft et al., 2014) and the Hindu Kush-Himalayan region (Schmid et al., 2015).

Google Earth contains the best freely available imagery for detecting rock glaciers across large spatial areas. The aerial identification and subsequent classification of rock glaciers in the Daxue Shan were supplemented with validation in the field where access permitted.

Rock glaciers are characterized by distinct flow features and structural patterns. Transversal or longitudinal flow features (ridges and furrows) are common on rock glaciers due to the deformation of their internal ice structures (Clark et al., 1998;

Humlum, 2000; Haeberli et al., 2006; Berthling, 2011). Many rock glaciers also exhibit structural patterns such as steep frontal slopes and side slopes with swollen bodies. Due to the constant supply of talus or debris, the surface textures of rock glaciers are usually different from those of the surrounding slopes. Depending on the mobility and permafrost presence, rock glaciers are usually divided into active, inactive, and relict rock glaciers three types (Sattler et al., 2016). In general, the presence of

ice within an active/inactive rock glaciers have a steep (>35°) frontal slope (Ikeda and Matsuoka, 2002) and a well-developed flow-like morphology defined by sets of parallel and curved ridges separated by long V-shaped furrows (Barsch, 1996; Roer and Nyenhuis, 2007), the absence or the sparse occurrence of vegetation (Onaca et al., 2013). Inactive rock glaciers also contain ice, but are immobile. In contrast, relict rock glaciers are characterised by surface collapse features as a result of permafrost

degradation, with gentler frontal and marginal slopes, and often vegetation cover (Wahrhaftig and Cox, 1959; Haeberli, 1985; Scotti et al., 2013). Based on these criteria, we visually examined the landforms found in the Google Earth images and identified any rock glaciers. We mapped the distribution of rock glaciers in the study region using the ASTER GDEM dataset (to within a horizontal accuracy of 30 m) and the Google Earth imagery, before marking the geographical location of each identified rock glacier and delineating its outline using Google Earth.

The topographical characteristics of the rock glaciers identified in the inventory were recorded in a GIS environment (ArcMap 10.2) and then extracted and recorded for each rock glacier; these characteristics were both qualitative and quantitative and included each rock glacier's geographical location (*i.e.*, the coordinates of its center), each rock glacier's type as determined using dynamic, genetic and geometric criteria (moraine-talus; tongue-lobate), and each rock glacier's aspect (north-facing, northeast-facing *etc.*), mean gradient of slope (°), area (km$^2$), centerline length (m), average width (m), average altitude (m

asl), debris source area (parameter) and bedrock lithology. A geological layer (using a geological map with a scale of 1:500,000 from the China Geological Survey) was added to the geographical location data for each rock glacier so that the relevant class of bedrock could be incorporated within the spatial distribution database.

Based on the main source of the mass input of debris into each rock glacier and its subsequent transport downslope, we subdivided rock glaciers into two distinct categories: talus-derived rock glaciers developing below talus slopes; and moraine-

20 type rock glaciers evolving mainly from glaciogenic materials (Lilleøren and Etzelmüller, 2016; Onaca et al., 2017) (Fig. 2). In terms of their planar geometry, these rock glaciers could be subdivided into two types: lobate and tongue-shaped (Fig. 2). The length/width ratio was used to distinguish between lobate (length/width ratio <1) and tongue-shaped (length/width ratio >1) rock glaciers (Giardino and Vick, 1987; Martin, 1987; Barsch, 1996; Guglielmin and Smiraglia, 1998; Onaca et al., 2017). The overall aspect of each rock glacier was manually derived for each feature according to the main direction of the rock glacier

flow before being recoded into eight categories which corresponded to the orientation of each rock glacier.

However, due to the lack of data regarding the flow behavior of rock glaciers, it remains to be determined whether these landforms are currently active, or whether they represent the fossilized remains of inactive rock glaciers; further analysis, when conditions permit, it therefore vital. In addition, some aspects of digitisation were challenging based on visual interpretation of remotely sensed imagery alone and thus inherently associated with uncertainty (Sattler et al., 2016; Jones et al., 2018b).

There are some rock glaciers may not be correctly delineated. Especially, delimitation of the upper boundary of rock glaciers through geomorphic mapping, is arbitrary (Krainer and Ribis, 2012); delineation of individual polygons where multiple rock glaciers coalesce into a single body, is inherently subjective (Scotti et al., 2013; Schmid et al., 2015). Moreover, several complex landforms may are delineated as rock glaciers which could also be landslide deposits or relict rock glaciers. Therefore, in the future research, adding additional data sources and geophysical field investigations would be necessary to further

increase the accuracy of the outlines of the rock glaciers. Further in situ observations would be useful to constrain methods of rock glacier identification and increase accuracy when building rock glacier inventories; such fieldwork would also supplement results rendered by the Digital Elevation Model (DEM) we used to determine the altitude and aspect of each rock glacier, and which we set to a 30 m spatial resolution. Further, a higher resolution DEM paired with in situ climate datasets would provide a more accurate representation of the distribution of the rock glaciers of the Daxue Shan. Due to the limitations imposed by

the 30 m spatial resolution and the uncertainties inherent in any artificial visual identification, we may have failed to identify all the rock glaciers of the Daxue Shan. These uncertainties explain why we chose to adopt a range of values rather than exact numerical figures during our statistical analyses of the formation and development of the rock glaciers of the Daxue Shan as controlled by local environmental factors.

### 3.2 Spatial and statistical analyses

We set the eight geographical and topographical parameters (*i.e.*, latitude, longitude, rock glacier (RG) area, length, width, altitude asl, mean gradient and aspect) for each of the rock glaciers of the Daxue Shan to an eight-dimensional random variable (*i.e.*, $X_1, X_2, X_3 ... X_8$). A correlation coefficient $\rho_{ij}$ (i, j = 1, 2 ... 8) of $X_i$ and $X_j$ was introduced into the correlation matrix of the random dimensional vector as an eight order matrix for each element, and was denoted by R, thus:

$$R = \begin{bmatrix} \rho_{11} & \rho_{12} & \cdots & \rho_{18} \\ \rho_{21} & \rho_{22} & \cdots & \rho_{28} \\ \vdots & \vdots & \vdots & \vdots \\ \rho_{81} & \rho_{82} & \cdots & \rho_{88} \end{bmatrix}, \rho_{ij} = \frac{cov(X_i, X_j)}{\sqrt{DX_i}\sqrt{DX_j}}, cov(X_i, X_j) = E\left((X_i - E(X_i)) \cdot \left(X_j - E(X_j)\right)\right)$$

The diagonal element of the correlation matrix was 1, and the correlation matrix itself was a symmetrical matrix. We performed the statistical analysis using SPSS20® software. Correlations between the quantitative topographical variables were evaluated using Pearson correlation coefficients at a corresponding significance level of $p<0.05$.

There may be collinearity between the terrain variables, and principal components analysis (PCA) can used to determine the relationships between them (White and Copland, 2015; Ran, 2017). However, in this study, we performed the KMO and Bartlett's Test with a KMO value of $0.387<0.5$ (Table 1), the original variable is not suitable for PCA, there is weak collinearity between the terrain variables. Therefore, in the case of convenient interpretation and calculation (not too many dimensions), without dimensionality reduction, the original variable information is retained as much as possible to obtain more terrain

information that affects the development of the rock glaciers.

**4 Results**

In total, 295 rock glaciers were identified in the Daxue Shan (Fig. 3), covering an area of 55.70 km$^2$ (Table 2). Of these, 149 of them (50.51%; total area 27.59 km$^2$) were talus-derived rock glaciers, the other 146 (49.49%; total area 28.11 km$^2$) were moraine-type rock glaciers; 279 (94.58%; total area 52.87 km$^2$) of the rock glaciers were tongue-shaped. The remaining 16

(5.42%; total area 2.83 km$^2$) were lobate-shaped. Most rock glaciers in the Daxue Shan are therefore tongue-shaped rock glaciers. In the study area, we also found that the number of rock glaciers on the southwest-facing slopes of Mt. Zheduo was significantly higher than on the southwest-facing slopes of the southwestern and northwestern sectors of the Daxue Shan. Although these two sectors and Mt. Zheduo exhibit similar environmental trends and receive solar radiation patterns, the higher altitudes asl of the southwest-facing slopes of Mt. Zheduo lead to lower temperatures than those observed for the northwestern

and southwestern sectors of the Daxue Shan. On the other hand, because the southwest-facing slopes of Mt. Zheduo are the most southwesterly of the whole mountain range, they experience higher levels of orogenic southwesterly monsoonal

precipitation (snowfall). This combination of factors makes the southwest-facing slopes of Mt. Zheduo more conducive to the development of periglacial landforms such as rock glaciers.

The 295 rock glaciers are found at altitudes of between 4,300 and 4,600 m asl, with the mean altitude being 4,471 m asl. Moraine-type rock glaciers are mainly concentrated in the 4,400~4,600 m asl zone, and talus-derived rock glaciers in the 4,300~4550 m asl belt. Tongue-shaped and lobate-shaped rock glaciers are mainly concentrated in the 4,350~4,600 m asl zone (Fig. 4a). We found that the asl altitudes of moraine-type rock glaciers were at least 50~100 m higher than for talus-derived rock glaciers. The upper boundaries for the vast majority of rock glacier types were ~4,600 m asl, at higher altitude there are often present some ice glaciers. Figure 4b shows the range in areas covered by different types of rock glaciers. Apart from a few outliers, it can be seen that the area of most rock glacier types area is <0.3 km$^2$, and that, in this regard, there is no clear difference between these different rock glacier types. Figure 4c shows the range in the mean gradients of the slopes of different types of rock glaciers. Moraine-type and talus-derived rock glaciers exhibit mean gradients which are all concentrated within the 22°~35° range. However, tongue-shaped and lobate rock glaciers display a greater difference in mean gradient. Tongue-shaped rock glaciers have slopes with mean gradients which are concentrated in the 22°~35° range, whereas the mean gradients of lobate rock glaciers fall within the 27°~45° range, meaning that the upper (~10°) and lower (~5°) slopes of tongue-shaped rock glaciers are both ~5° lower than for lobate rock glaciers. Figure 4d displays the range in the lengths of different types of rock glaciers. Moraine-type, talus- type and tongue-shaped rock glaciers are mostly 500~1000 m long, whereas lobate rock glaciers are mostly 200~400 m long, compared with lobate rock glaciers, moraine-type and tongue-shaped rock glaciers have more sediment supplies and last longer on gentle slope, indicating that moraine-type and tongue-shaped rock glaciers flow further than lobate rock glaciers.

Our dataset revealed that, apart from south-facing (5.44%), southeast-facing (3.06%) and northeast-facing (20.75%) slopes, the rock glaciers of the Daxue Shan are fairly evenly distributed on slopes with the remaining five aspects, which each aspect accounting for ~15% of the total. Moraine-type rock glaciers are most often northeast-facing (30.34%) and north-facing (20%), but talus-derived rock glaciers are most often southwest-facing (22.82%) and west-facing (17.45%); they are less commonly southeast-facing (5.37%), south-facing (6.71%) and north-facing (8.72%). Lobate rock glaciers tend to be found less on south-facing (6.25%) and southeast-facing (0%) slopes, but more commonly on north-facing, northwest-facing and east-facing,

which each aspect accounting for ~18.75% of the total. We compared all our results and discovered that shady (*i.e.*, N, NE and E) slopes appear more conducive to the formation of moraine-type rock glaciers, and sunny (*i.e.*, W, SW and S) slopes appear more conducive to the formation of talus-derived rock glaciers. In addition, there are more steep rock walls on the north faces producing debris, north-facing (*i.e.*, N, NW and NE) slopes seem to be more favorable for the formation of lobate rock glaciers

than do south-facing (*i.e.*, SW, S and SE) ones (Fig. 5).

The mean altitude of a rock glacier's front (MAF) has often been taken to be a good approximation of the lower boundary of the discontinuous permafrost zone (*i.e.*, Scotti et al., 2013). We found a significant altitudinal difference between the lower permafrost boundaries identified on the abovementioned eight aspects as they were categorized for the Daxue Shan. For example, permafrost was assumed to be probable above 4,300 m asl on east-facing slopes, and above 4,403 m asl on west-

facing slopes. The mean lower permafrost boundary was calculated as occurring at 4,352 m asl (derived from a mean value of 4,315 m asl for east-facing slopes at 4315m, and 4,419 m asl for west-facing slopes). The mean lower permafrost boundary on east-facing (shady) slopes would therefore probable be 104 m lower than that of west-facing (sunny) slopes (Fig. 6).

**5 Discussion**

The spatial distribution and dynamics of rock glaciers are especially dependent upon the local topography and climate

(Springman et al., 2012; Delaloye et al., 2013). Analyzing local environmental factors is therefore crucial to obtaining an understanding of the formation, development and spatial distribution of rock glaciers.

**5.1 Topographical controls on rock glaciers**

The results showed that there is a significantly positive correlation (*p*=0.05) between latitude, altitude asl and rock glaciers length (Table 3), a relation which is locally determined by the topographical characteristics of the Daxue Shan. With the

increase of latitude from the south to the north in the Daxue Shan, the high altitude rock glaciers increase, and flow further downvalley than low altitude, these topographical characteristics result in the rock glaciers altitude asl and length increase with latitude. The altitudes of the mountains and length increase with latitude along with a latitudinal decrease in air temperatures, meaning that the northern sector of the Daxue Shan has an environment which is more conducive to the formation of rock

glaciers and other periglacial landforms. Likewise, there is a significantly negative correlation ($p$=0.01) between latitude and longitude, indicating no significant impact upon the NW-SE clusters of rock glaciers found in the Daxue Shan region. There is also a significantly negative correlation ($p$=0.01) between longitude and altitude. The lower altitude areas to the east are less conducive to the development of rock glaciers, with the increase of longitude and the decrease of altitude, the closer it is to warm and humid, which kind of climatic conditions are not conducive to the formation of permafrost landforms such as rock glaciers. A significantly negative correlation ($p$=0.01) exists between rock glacier length and mean gradient of slope; the shortest rock glaciers are the talus-derived variety, and these have usually developed in steep topographical environments. Rock glacier area and mean gradient of slope have a significantly negative correlation ($p$=0.01); the larger rock glaciers are mostly concentrated on gentle slopes, meaning that they are more conducive to the development of large rock glaciers. In summary, the topography of the Daxue Shan is an important environmental control on the formation, development and spatial distribution of the region's rock glaciers.

In addition, the formation and development of the rock glaciers of the Daxue Shan are also strongly influenced by the landforms created by glacial erosion and deposition. The southeastern margins of the TP (where the Daxue Shan are located) are in a region of Quaternary glaciation which has been, and continues to be, strongly affected by monsoonal atmospheric circulations (Owen et al., 2005). This region possesses numerous ancient glacial relics and abundant landforms created by glacial erosion and deposition (Li and Yao, 1987). We found that the distribution of rock glaciers in close association with ice glaciers in the Daxue Shan, the upper boundaries for rock glaciers were ~4,600 m asl, at higher altitude there are often present some ice glaciers. In the context of global warming, it is widely accepted that the majority of glaciers on the Tibetan Plateau (TP) and its surroundings have experienced accelerated reduction (Bolch et al., 2012; Yao et al., 2012). The rate of glacier decline in Daxue Shan was -0.25 ± 0.20% $a^{-1}$ during 1990-2014 (Wang et al., 2017), some ice glaciers transforming to rock glaciers. Glacial depositional landforms (*e.g.*, moraine ridges) are highly conducive to the formation and development of moraine-type rock glaciers. Moraine ridges or moraines left after the retreat of the ancient glaciers can provide significant quantities of boulders, erratic blocks, debris, sand and ground ice. In the process of downward peristalsis, rock glaciers can incorporate old moraine material as well as the debris from both sides of the moraine ridge. Glacial erosional landforms in particular evince a closely relation with the formation and development of talus-derived rock glaciers. Ice structures, snow layers and moraines

within glaciers collapse from time to time, supplying talus to the feet of mountains. As a result of the freeze-thaw process and the effect of gravity, talus creep then forms rock glaciers.

## 5.2 Climatic controls on rock glaciers

The west-facing slopes of the Daxue Shan lie in the intersection between a sub-frigid monsoonal and a continental plateau climatic zone, and therefore experience a cold-temperate climate. At the Daofu meteorological station (2,957.2 m asl), mean annual precipitation (MAP) is ~613.5 mm, and mean annual temperature (MAT) is ~8.14°C (Fig. 7b). Based on an adiabatic rate of 0.65°C/100 m, we estimated the MAT at 4,311 m asl (*i.e.*, the lower permafrost boundary) to be ~-0.66°C. The east-facing slopes of the Daxue Shan are affected by a subtropical monsoonal climatic environment, and are affected principally by a southwesterly monsoonal atmospheric circulation, but also by a southwesterly monsoonal atmospheric circulation, and by the Westerlies. These slopes therefore experience high levels of precipitation (snowfall). MAP at the Kangding meteorological station (2,615.7 m asl) reaches 858.3 mm; MAT is ~7.29 °C (Fig. 7a). We calculated the MAT at 4,352 m asl (*i.e.*, the lower permafrost boundary) to be ~-4.00°C. Here, the freeze-thaw process would be frequent (Fig. 7), meaning that the climatic environment would provide temperature and precipitation conditions highly favorable to the formation and development of rock glaciers.

The cryosphere reacts sensitively to climate change (Gruber et al., 2017). Compared with Gruber's (2012) global Permafrost Zonation Index (PZI) map, the rock glaciers distribution in the Daxue Shan is in good agreement with the PZI on the whole and some rock glaciers are situated within the PZI fringe of uncertainty (Fig. 3). Strictly controlled by the temperature decreasing with increasing altitude, further indicating the climatic controls on development of permafrost such as rock glaciers. In addition, compared with the distribution of glaciers in the Daxue Shan (Wang et al., 2017), the distributions of rock glaciers also has the characteristics of small differences between the south and north, owing to a north–south corridor effect for water and heat transport and diffusion through the longitudinal gorges in the Daxue Shan. It is the result of climatic and topographical comprehensive control on rock glaciers.

## 5.3 Lithological controls on rock glaciers

Lithology is a critical control for the supply of talus to ice- and rock-glacier surfaces (Haeberli et al., 2006). Figure 8 shows that the major exposed strata in the Daxue Shan region are composed of Tertiary monzonitic granite, consistent with the NW-SE trending Xianshuihe Fault. The surrounding mountains in this area generally consist of biotite-muscovite granite that intruded 16~13 Ma ago (Roger et al., 1995). Also located in this region is the tectonically important Zheduotang Fault, which runs through the Zheduo Valley, and is one of the most active fault systems on the TP's margins (Allen et al., 1991). It can be seen from Figure 8 that the distribution of rock masses along the Xianshuihe Fault in the Daxue Shan region is clearly controlled by this NW-SE left-lateral strike-slip fault.

In contrast to other regions (Lilleøren and Etzelmüller, 2016; Onaca et al., 2017), we found that in the Daxue Shan both moraine-type and talus-derived rock glaciers have developed in the monzogranitic areas, and that rock glacier and monzonitic granite exhibit a high spatial correlation and interdependence. The Tertiary monzogranites of the Daxue Shan are clearly highly conducive to the formation and development of rock glaciers. This is consistent with the findings of Onaca et al. (2017) in the southern Carpathian Mountains. According to Popescu et al. (2015), rock glaciers located in granitic and granodioritic massifs are composed of larger clasts compared with those found in metamorphic massifs. Thus, the higher porosity of the substrata in granitic and granodioritic massifs allows for a significant cooling beneath the bouldery mantle because the denser cold air is trapped between the large boulders (Balch, 1900). The lithological and mineralogical characteristics which accompany the high porosity of tertiary monzogranites are therefore more favorable to the formation and development of local rock glaciers than are other lithologies. In addition, rock glacier formation also controlled by slope and sedimentation rates contributing debris to the landforms (Müller et al., 2016). There are a large sources of sediment and sediment storages in the Daxue Shan, and are controlled by the processes occurring within this setting (Müller et al., 2014). An abundance of steep rock walls and deepened valley sides, provides catchment areas for rock glacier development, combined with intense monsoonal precipitation and tectonic activity, drives sediment transport processes and rock glacier development in the Daxue Shan.

Several researchers (*e.g.*, Cui and Zhu, 1989; Zhu, 1992; Zhu et al., 1992; Liu et al., 1995; Bolch and Gorbunov, 2014) have previously identified hundreds of rock glaciers in the northern Tianshan Mountains. They found that most of the identified

rock glaciers were tongue-shaped, and were located at altitudes between 3,300 and 3,900 m asl, on north-facing slopes. Most rock glaciers in the Daxue Shan are also tongue-shaped. However, the altitudes at, and the aspects on, which these rock glaciers are found differ between the Daxue and the Tianshan mountain ranges. First, in terms of altitude, the rock glaciers of the Daxue Shan are located at altitudes between 4,300 and 4,600 m asl, higher than the Tianshan rock glaciers by approximately 700~1000

5   m. It would be reasonable to assume, therefore, that the rock glaciers located in lower latitudes are more likely to be found at higher altitudes. Second, in terms of aspect, the rock glaciers of the Daxue Shan are more evenly distributed across all eight abovementioned aspects than are the rock glaciers of the Tianshan Mountains. This could be explained by several factors, including the differences in overall altitude, as well as in the orientation of the main massif of each mountain range. The Daxue Shan lie along an approximately NW-SE axis, whereas the Tianshan Mountains are roughly W-E in presentation. Rock glaciers

are therefore less commonly found on the east- and west-facing slopes of the Tianshan. The effect of solar radiation is stronger on the south-facing slopes of the Tianshan Mountains than on its north-facing ones, meaning that conditions on these south-facing slopes are less conducive to the development of rock glaciers; most of the range's rock glaciers are therefore found on its north-facing slopes. Furthermore, when higher altitudes are reached, all aspects experience lower air temperatures, resulting in a lessening of the impact caused by the difference between air temperature and solar radiation exposure; this phenomenon

is similar to that found in the Daxue Shan, and explains why rock glaciers there are fairly evenly distributed on all eight aspects. However, when altitudes are lower, the impact of solar radiation, combined with warmer air temperatures, is greater, particularly on south-facing slopes; both temperature and solar radiation are lesser on shady north-facing slopes, however, explaining the predominance of north-facing rock glaciers in the Tianshan Mountains.

**6 Conclusions**

Rock glaciers are widespread in the Daxue Shan; of these, tongue-shaped rock glaciers cover the largest area. The occurrence and characteristics of these rock glaciers can mostly be explained by local environmental controls.

In total, 295 rock glaciers were identified in the Daxue Shan, covering a total area of 55.70 km$^2$. The altitudes at which moraine-type rock glaciers are found (*i.e.*, 4,400~4,600 m asl) are at least 50~100 m higher than for talus-derived rock glaciers (*i.e.*,

4,300~4,550 m asl), although the upper altitudinal limit for both these types of rock glacier is ~4,600 m asl, at higher altitude there are often present some ice glaciers. Except for a few outliers, the area of each type of rock glacier is no greater than 0.3 km$^2$. There is no significant difference between moraine-type and talus-derived rock glaciers in terms of the mean gradients of the slopes upon which the glaciers are found (*i.e.*, they are all clustered within the 22~35° range), but the upper and lower

mean slope gradients of tongue-shaped rock glaciers are ~10° and ~5° lower than for lobate rock glaciers respectively. Moraine-type, talus-derived and tongue-shaped rock glaciers are longer (*i.e.*, 500~1000 m) than lobate rock glaciers (*i.e.*, 200~400 m). We found shady (*i.e.*, N, NE and E) slopes more conducive to the formation of moraine-type rock glaciers than sunny (*i.e.*, W, SW and S) ones, while sunny (*i.e.*, W, SW and S) slopes appear more conducive to the formation of talus-derived rock glaciers. In addition, north-facing (*i.e.*, N, NW and NE) slopes appeared more favorable to the formation of lobate

rock glaciers than did south-facing (*i.e.*, SW, S and SE) ones. The mean regional lowest altitudinal limit of rock glaciers is 4,352 m asl, an altitude which was taken to indicate the local permafrost's mean lower boundary. On east-facing slopes, the permafrost's lower boundary can therefore reasonably be assumed to be 104 m lower than on west-facing slopes.

Environmental controls (*i.e.*, topographical, climatic and lithological factors) play a very important role in the formation and development of the rock glaciers of the Daxue Shan. The correlation matrix of rock glacier parameters indicates that the

formation of rock glaciers is closely related to local topographical parameters. The local climatic environment leads to a frequent freeze-thaw process within these rock glaciers, a process which is also beneficial to their formation and development. Tertiary monzonitic granite, with its large clastic and highly porous characteristics, is more sensitive than other lithological components to the freeze-thaw process, and continuous weathering of this monzogranitic substratum thus provides the ideal raw material for the rock glaciers of the Daxue Shan.

**Data availability**

The data associated with this article can be found in the Supplement. These data include Google maps of the most important areas described in this article, as well as a tabulation of the parameters of the rock glaciers found in the Daxue Shan.

**Competing interests**

The authors declare no competing interests, financial or otherwise.

**Acknowledgements**

This work was funded by the National Natural Science Foundation of China (Grant Nos. 41230743 and 41371082). We should like to express our appreciation to the people who have revised this article and for the great interest they have taken in improving it.

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

**Tables and Figures:**

**Table 1. KMO and Bartlett's Test.**

| Kaiser-Meyer-Olkin Measure of Sampling Adequacy. | | .387 |
|---|---|---|
| Bartlett's Test of Sphericity | Approx. Chi-Square | 1216.315 |
| | df | 28 |
| | Sig. | .000 |

**Table 2. Statistics for the 295 rock glaciers found in the Daxue Shan.**

| RG type | Number of landforms | RG area (km²) | Altitude (m asl) | Length (m) | Width (m) | Gradient of Slope (°) | MAF (m asl) |
|---|---|---|---|---|---|---|---|
| Moraine | 146 | 28.11 | 4,501 | 793 | 235 | 28.45 | 4,385 |
| Talus | 149 | 27.59 | 4,442 | 805 | 228 | 30.05 | 4,321 |
| Tongue | 279 | 52.87 | 4,470 | 829 | 211 | 28.89 | 4,347 |
| Lobate | 16 | 2.83 | 4,491 | 275 | 582 | 35.69 | 4,447 |
| MTRG | 139 | 26.86 | 4,496 | 817 | 218 | 27.96 | 4,377 |
| MLRG | 7 | 1.25 | 4,592 | 303 | 564 | 38.29 | 4,539 |
| TTRG | 140 | 26.01 | 4,444 | 841 | 204 | 29.81 | 4,317 |
| TLRG | 9 | 1.58 | 4,412 | 253 | 595 | 33.67 | 4,376 |
| All RG | 295 | 55.70 | 4,471 | 799 | 231 | 29.26 | 4,352 |

Note: RG=rock glaciers; MTRG= moraine-type and tongue-shaped rock glaciers; MLRG= moraine-type and lobate rock glaciers; TTRG=talus-derived and tongue-shaped rock glaciers; TLRG= talus-derived and lobate rock glaciers; MAF= minimum altitude of rock glacier front. Altitude of rock glacier, altitude of rock glacier front, length, width and gradient of slope are all mean values.

**Table 3. Correlation matrix of rock glacier parameters; marked correlations (bold) are significant at the significance level of *p*=0.01 (\*\*) and *p*=0.05 (\*).**

|  | Latitude | Longitude | Altitude | Length | Width | RG area | Mean slope | Aspect |
|---|---|---|---|---|---|---|---|---|
| Latitude | 1.000 | **-0.893\*\*** | **0.116\*** | **0.102\*** | -0.020 | 0.029 | 0.092 | -0.016 |
| Longitude | **-0.893\*\*** | 1.000 | **-0.290\*\*** | -0.062 | 0.025 | 0.002 | -0.004 | -0.034 |
| Altitude | **0.116\*** | **-0.290\*\*** | 1.000 | -0.075 | 0.087 | 0.031 | **-0.102\*** | 0.045 |
| Length | **0.102\*** | -0.062 | -0.075 | 1.000 | 0.063 | **0.776\*\*** | **-0.341\*\*** | 0.013 |
| Width | -0.020 | 0.025 | 0.087 | 0.063 | 1.000 | **0.572\*\*** | -0.004 | -0.026 |
| RG area | 0.029 | 0.002 | 0.031 | **0.776\*\*** | **0.572\*\*** | 1.000 | **-0.265\*\*** | 0.010 |
| Mean slope | 0.092 | -0.004 | **-0.102\*** | **-0.341\*\*** | -0.004 | **-0.265\*\*** | 1.000 | -0.068 |
| Aspect | -0.016 | -0.034 | 0.045 | 0.013 | -0.026 | 0.010 | -0.068 | 1.000 |

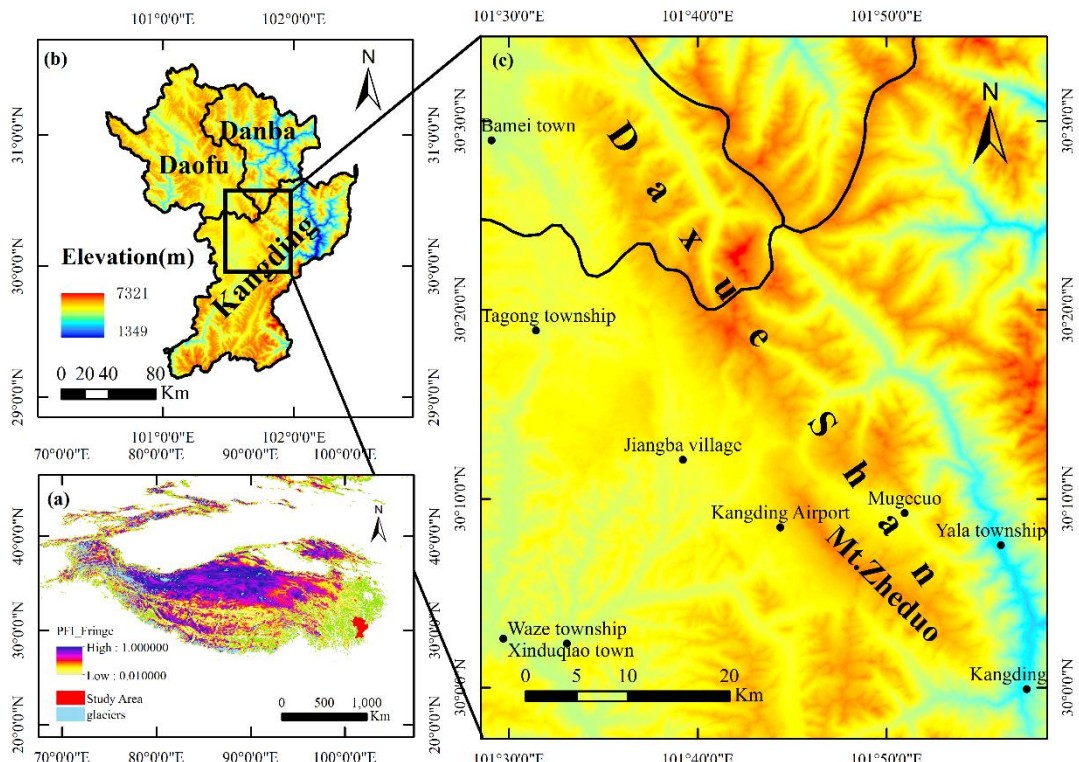

**Figure 1: (a) The location of the study area in the permafrost zone of the TP. The Permafrost Zonation Index (PZI), or a corresponding map color, indicates to what degree permafrost exists only under the most favorable conditions (yellow), or nearly everywhere (blue); the map was produced using a temporal resolution of 30 arc-seconds (<1km) on a WGS84 lat/lon grid (Gruber, 2012). (b) and (c) are the geographical and topographical maps of the study area based on a spatial resolution of 30 m using ASTER-GDEM v2 software, as shown in the WGS84 coordinate system.**

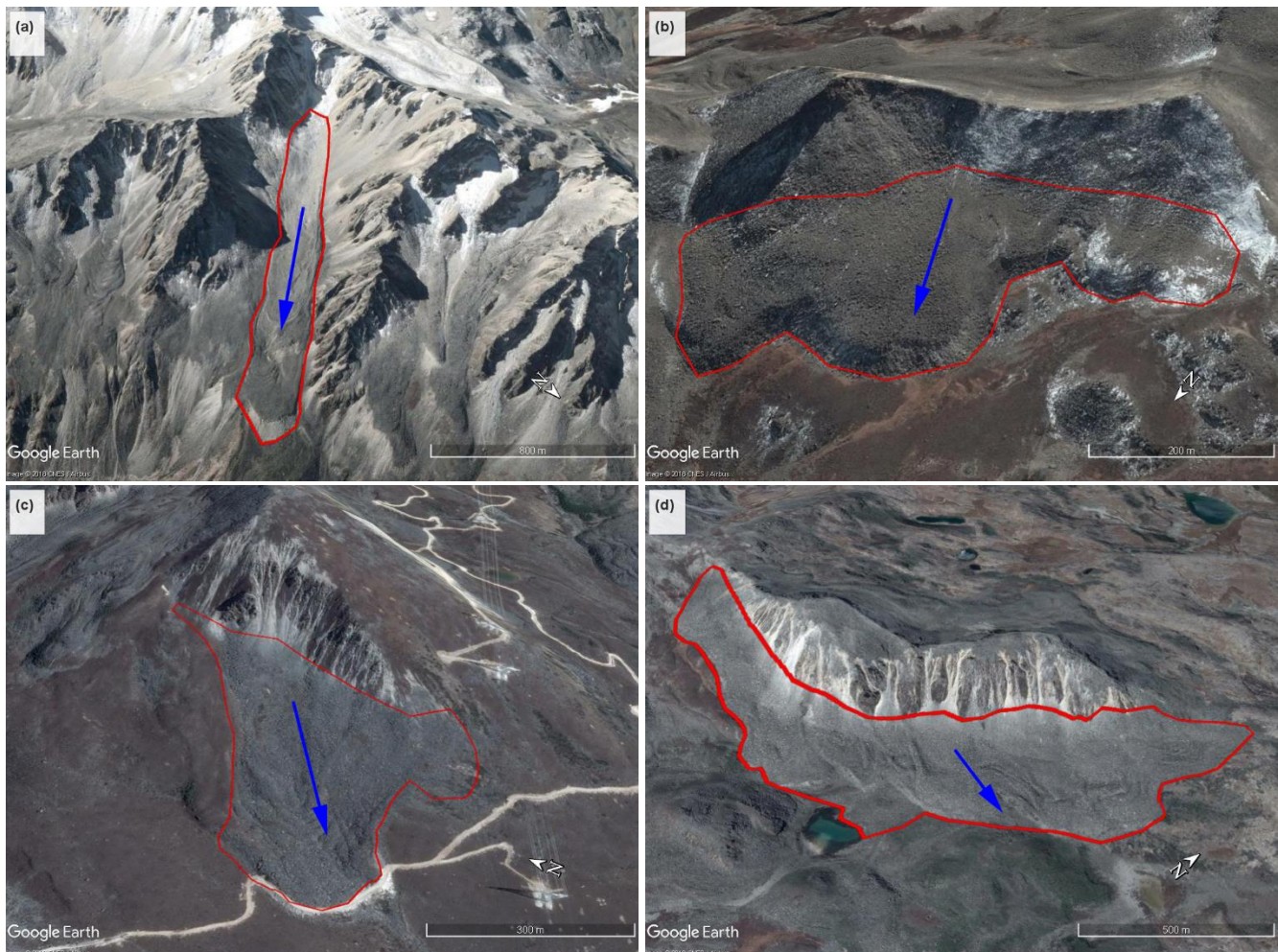

**Figure 2: Examples of different types of rock glaciers in the Daxue Shan: (a) moraine-type and tongue-shaped rock glaciers (30.332767ºN, 101.707756ºE) (30th January, 2017); (b) moraine-type and lobate rock glaciers (30.217147ºN,101.791585ºE) (15th November, 2015); (c) talus-derived and tongue-shaped rock glaciers (30.067066ºN, 101.819432ºE) (21st October, 2014); (d) talus-derived and lobate rock glaciers (30.127825ºN, 101.812158ºE) (21st October, 2014). The red lines show the outlines of the rock glaciers; the blue arrows indicate the direction of flow of the rock glaciers. Source: Google Earth.**

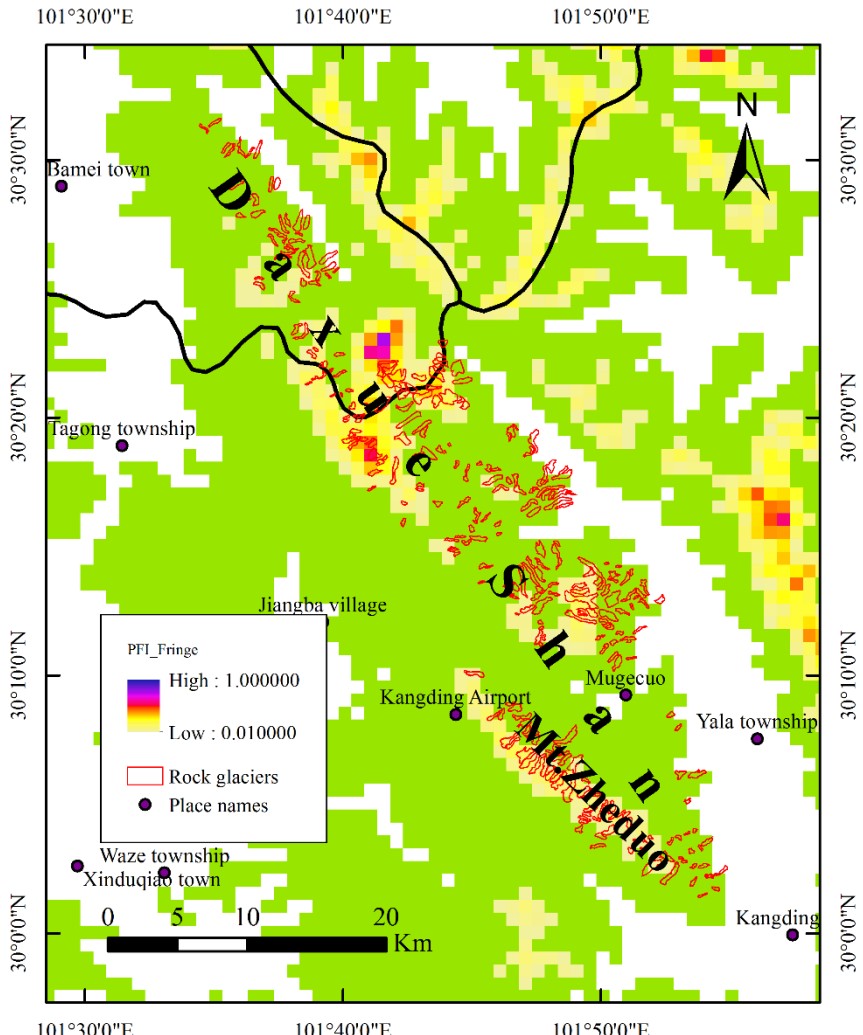

**Figure 3: Spatial distribution of rock glaciers and Permafrost Zonation Index (PZI) in the Daxue Shan. The PZI data sources: Gruber's (2012), the green area represent the fringe of uncertainty.**

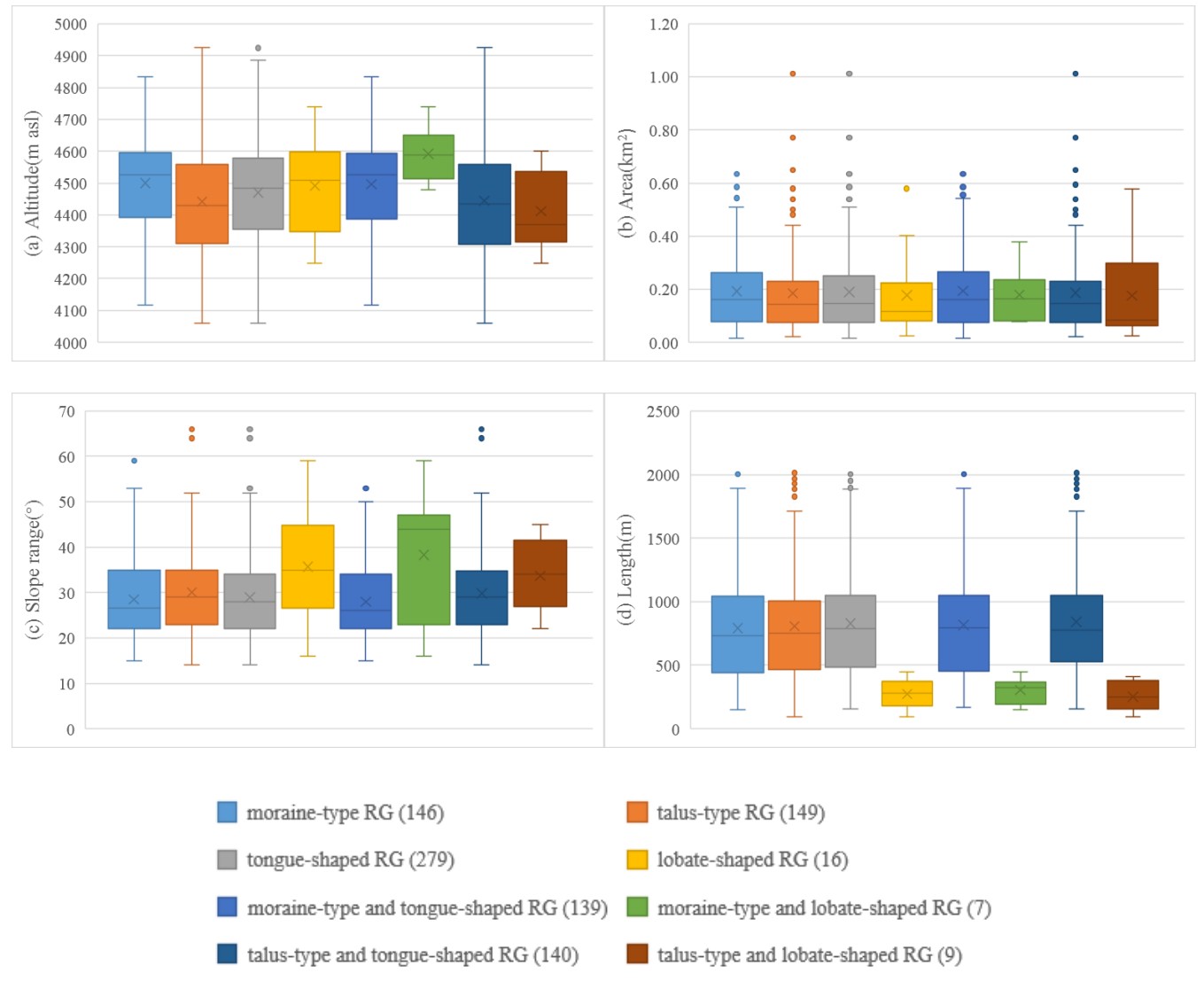

**Figure 4: Boxplots illustrating the distributional characteristics of rock glaciers in the Daxue Shan: (a) average altitude (m asl); (b) area (km²); (c) range in the gradient of the slope (°); and (d) length (m). Boxplots represent 25-75% of all values, the caps at the ends of the vertical lines represent 10-90% of values, and the line in the center of each box indicates the median value. The number of the population of the different kinds of rock glacier is in the brackets of the legend.**

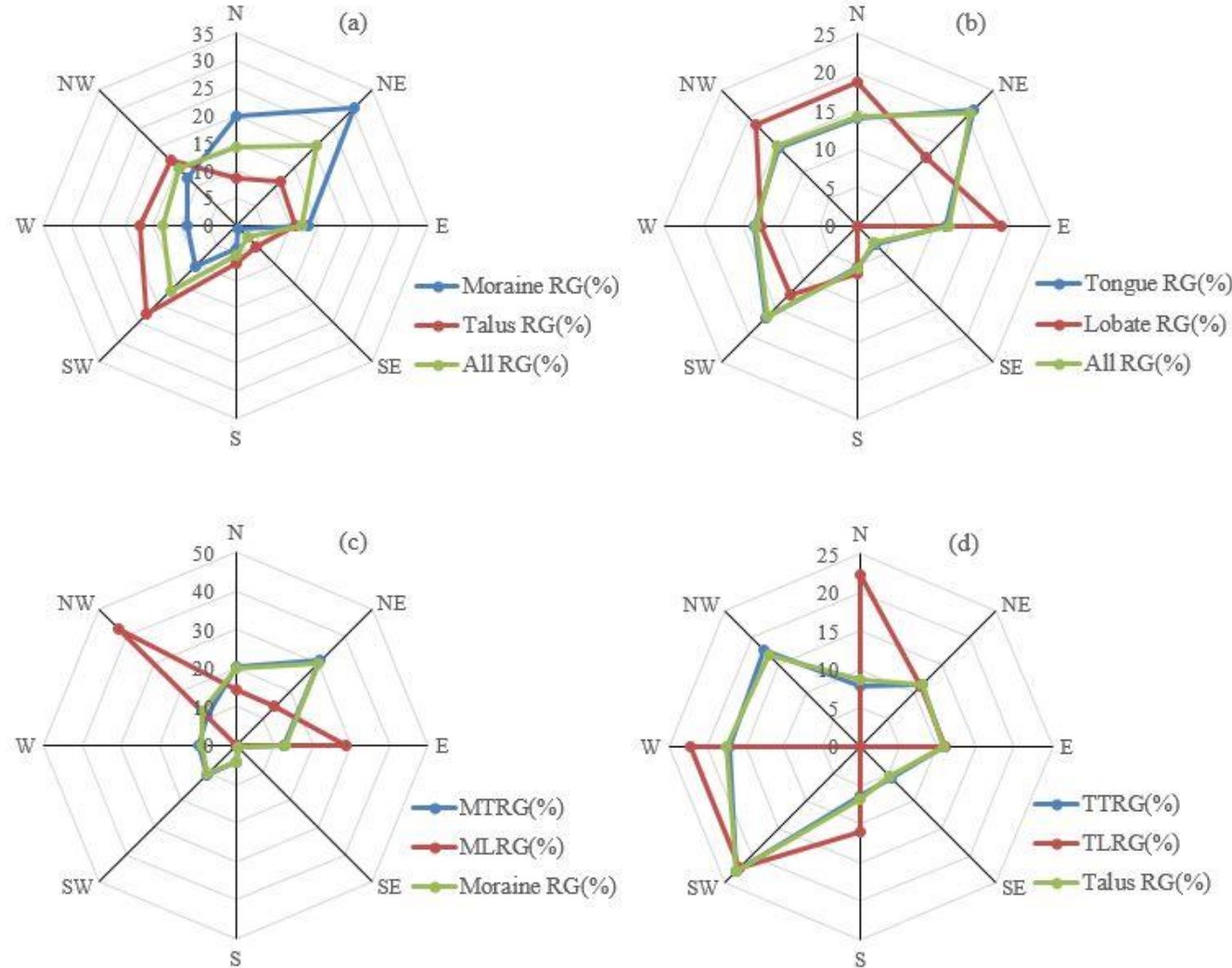

**Figure 5: Analysis of the abundances of different rock glacier types versus aspect. The number of rock glaciers for each aspect on each of the four radar plots is shown as a percentage (%). (Note: RG=rock glaciers; MTRG= moraine-type and tongue-shaped rock glaciers; MLRG= moraine-type and lobate rock glaciers; TTRG=talus-derived and tongue-shaped rock glaciers; TLRG= talus-derived and lobate rock glaciers)**

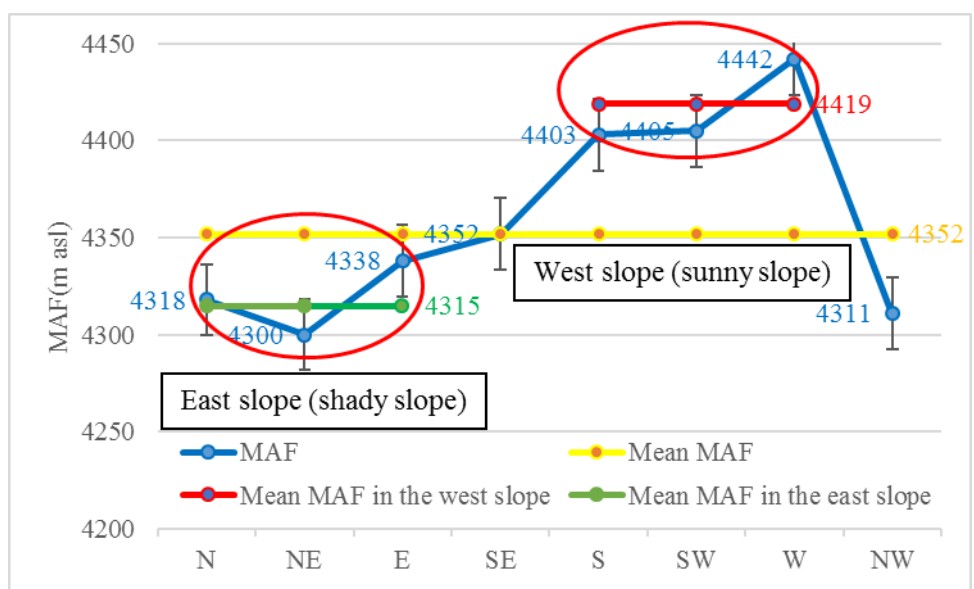

**Figure 6: Minimum altitudinal rock glacier fronts (MAF) for all eight aspects, along with the overall mean. These values are taken to represent the lower boundaries of the potential permafrost extent in the Daxueshan region (bars indicate standard errors of the mean). Because the Daxue Shan lie along an approximately NW-SE axis, we used this NW-SE axis as the boundary separating east-facing (*i.e.*, N, NE, E), shady slopes from west-facing (*i.e.*, S, SW, W), sunny slopes.**

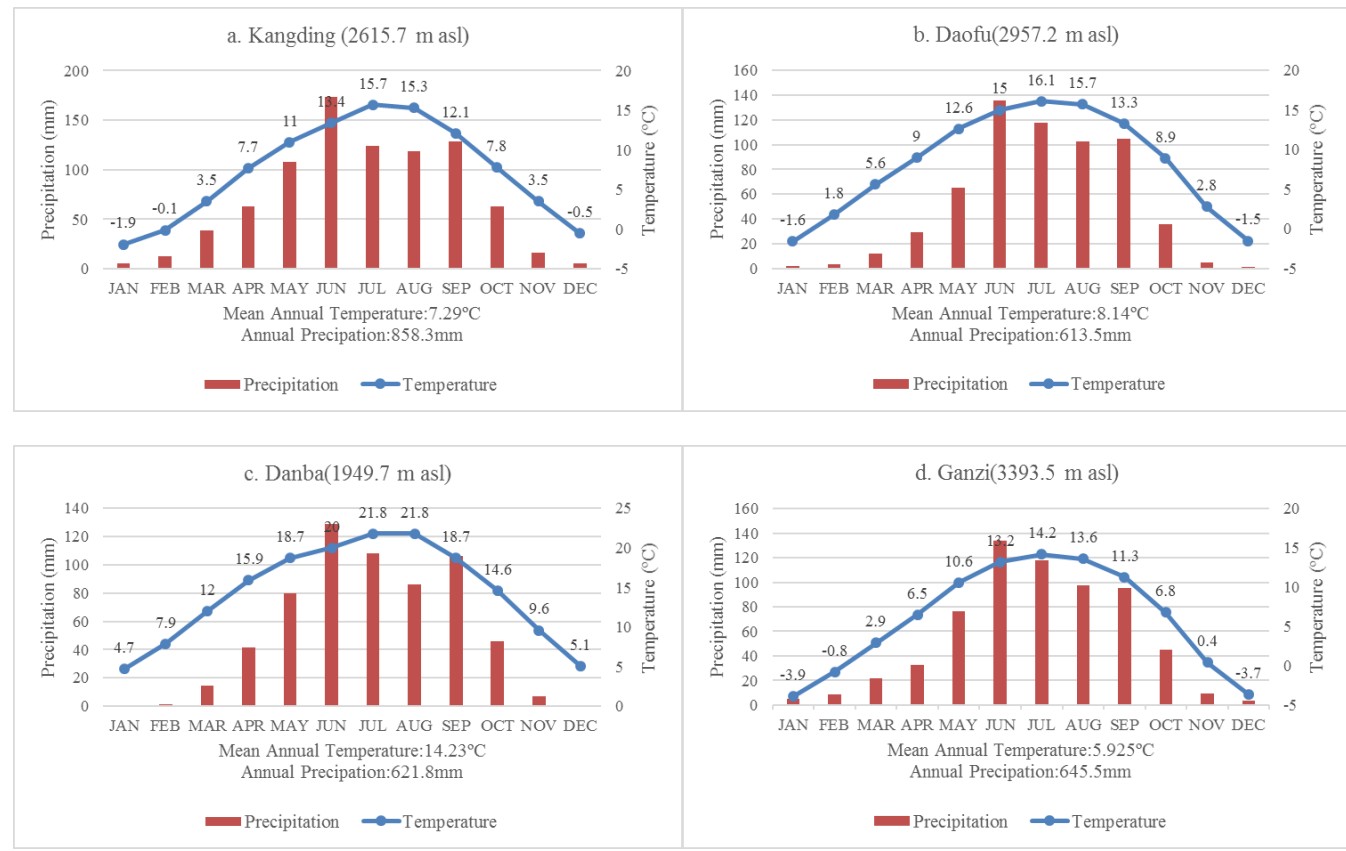

**Figure 7: Climatographs for the Kangding (2,615.7 m asl, 30.03°N, 101.58°E), Daofu (2,957.2 m asl, 30.59°N, 101.07°E), Danba (1,949.7 m asl, 30.53°N, 101.53°E) and Ganzi (3,393.5 m asl, 31.37°N, 100°E) meteorological stations. Data sources: Meteorological Data Center of the China Meteorological Administration (calculated for the period 1981–2010, inclusive).**

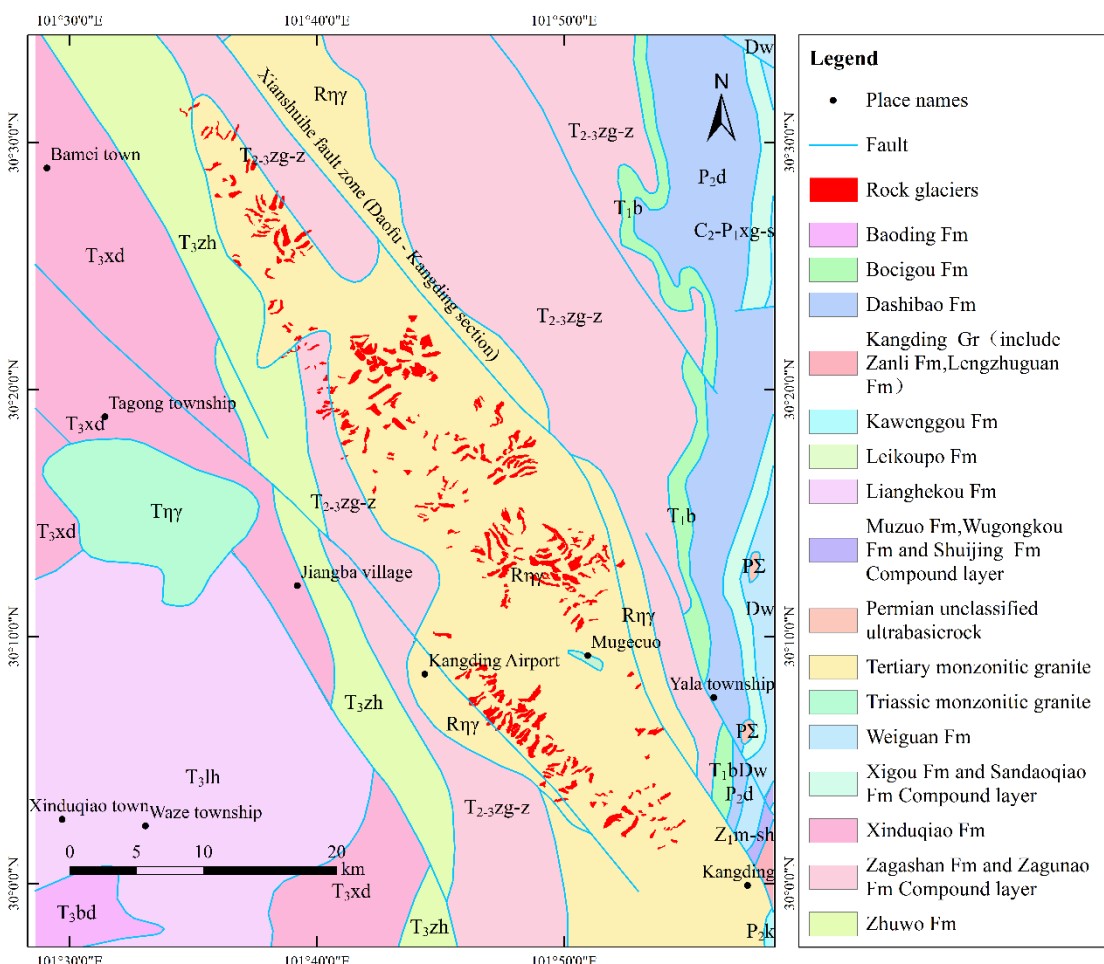

**Figure 8: The rock glaciers of the Daxue Shan rock glaciers superimposed on the local lithologic-geologic environment (lithological map reconstructed from a 1:500,000-scale digital geological map).**