# Peer review of "Rock glaciers in the Daxue Shan, southeastern Tibetan Plateau: an inventory, their distribution, and their environmental controls"

_The Cryosphere, 2017_

## Referee Comment (RC1) · J. Müller (Referee) · 12 Mar 2018

The authors introduce a novel rock glacier inventory of the Daxue Shan mountain range in the southeastern Tibetan Plateau. They use Google earth imagery to visually identify and map rock glaciers in the entire area. Summplementary data such as the ASTER GDEM and lithological information are implemented to assign localized geomporho-metric and subsurface attributes which is used for quantitative and qualitative analysis. The methods applied in this manuscript are well established and the analysis also doesnt hold any surprises but it is still a novel dataset presenting the rock glacier occurrence and distribution in the south eastern Tibetan Plateau. It is overall a further step towards a global rock glacier map. I therefore recommend the publication of this manuscript after moderate revisions. Please finde some remarks in the following and very specific comments in the attached pdf where i implemented some comments.

Specific comments : Methods P5L2 You need to elaborate more on the topographic specifications of active, inacitive and fossil rock glaciers. Bc with your approach it is hard to idnetify between the three but there are certain proxies such as subsidence and vegetation which can be used to determinte the state of the RG You mention in the abstract that you also use field data for the analysis but you never mention what kind of field data you acquired and how you use dit. You mention environmental controls like temperature and temperature dynamics like freeze thaw cycles numerous times in the manuscript but you never show any data. Maybe you have access to some high mountain temperature data in the area which you can show and help you with your argument. Not just the annual means as table 3 but also the annual or multiannual dynamics

4.Results and Discussion Since you manually derived the RG geomoetries it would be great if you could elaborate on the accuracy of your method. Did you have several persons working on the digitization of the RGs and did they perform differently or do you have more accurate field data which you could compare to the manual mapping and are there any differences ? I would suggest to refrain from using latitude and longitude to analyse RG properties since lat and long do not describe any environmental parameter but rather the regional topographical setting is more important. And thats the parameter that changes with Lat and Long. Focus more on the regional settings such as aspects, debris sources and valley/slope oreintation to interpret RG properties. It would be very beneficial if you include a descritption of the topographical characteristics of the study site in relation to the formation and evolution of rock glaciers. This would also help to understand the spatial setting which goes with the latitudinal impact.

4.2.3 Lithological controls on rock glaciers The lithological setting influences RG formation mainly by steepness and sedimentation rates contributing debris to the landforms. Please include this aspect into your elaboration and cite some references supporting the influence of lithology towards RG formation and evolution.

Also you mention the existence and application of g in-situ ground truthing data but you never explain how, where and what kind of data you gathered and used. Please include this either in the method or discussion section.

References Morris, S. E.: Topoclimatic Factors and the Development of Rock Glacier Facies, Sangre de Cristo Mountains, Southern Colorado, Arct. Alp. Res., 13, 329, doi:10.2307/1551039, 19

Wahrhaftig, C. and COX, A.: Rock Glaciers in the Alaska Range, Geol. Soc. Am. Bull., 70, 383, doi:10.1130/0016- 7606(1959)70[383:RGITAR]2.0.CO;2, 1959

Please also note the supplement to this comment:
https://www.the-cryosphere-discuss.net/tc-2017-290/tc-2017-290-RC1-supplement.pdf

---

## Referee Comment (RC2) · T. Bolch (Referee) · 17 Apr 2018

Review on the manuscript entitled

"Rock glaciers in the Daxue Shan, southeastern Tibetan Plateau: an inventory, their distribution, and their environmental controls"

by Zeze Ran and Gengnian Liu

*General comments:*

The study presents the first comprehensive rock glacier inventory of a specific mountain range in the southeastern Tibetan Plateau. Moreover different controls of their occurrence are analysed and described. Currently only little is known about rock glacier occurrence in Tibet and no comprehensive study exists. The topic of the study is therefore relevant and suitable for The Cryosphere. The text is, however, quite descriptive and needs moreover some structural and language improvements. I appreciate very much that the generated inventory is provided as supplementary material to this article. The major shortcoming is that the generated rock glacier inventory contains many errors and needs to be completely revised. Moreover, an analysis or a discussion of uncertainty is entirely missing. More details are provided below.

*Most critical issue:*

The authors delineate rock glaciers manually by on screen digitising based on high resolution satellite images available on Google Earth. General characteristics of rock glaciers are described based on the literature which the authors took as the basis for their delineation. However, from figures 2, 3, and 7 and from the provided vector data it is obvious that many rock glaciers are not correctly delineated. Especially the upper boundary of the rock glaciers is often wrong. Moreover, several complex landforms are delineated as rock glaciers which could also be landslide deposits or relict rock glaciers. I provide below two examples below: In both complex landforms are combined into one large rock glacier, but parts of the area are very probably not part of creeping permafrost bodies. Moreover, steep headwalls are also classified as rock glaciers which deliver debris, but are certainly not part of rock glaciers. In addition, some of the delineated features may not be rock glaciers. I am aware that it is partly impossible to clearly identify rock glaciers based on optical imagery alone. However, more effort is needed to correctly classify rock glaciers and hence, the rock glacier inventory needs to be completely revised at best using additional data sources. The newly available High Mountain Asia DEM, information providing information about the surface displacement (e.g. SAR coherence images) and the occurrence of permafrost (e.g. based on the Permafrost Zonation Index (Gruber, 2012, TC) or Chinese permafrost maps) will be important additional data sources. I am aware that this requires huge additional effort, but it is important that the inventory is as accurate as possible in order not to provide a wrong example about how to delineate rock glaciers and in order to not provide wrong numbers. Alternatively the authors may also revise their inventory using at least some additional data, only include intact (active and inactive rock glaciers) into the inventory, and classify them according to the certainty.

[Figure]

[Figure]

**Figures: Random examples of "rock glacier" outlines based on the kml-file provided by the authors.**

*Further major comments:*

1. The description of the methodology and the criteria for identification rock glaciers needs to be much more explicitly mentioned and described considering the relevant literature. In addition, an analysis and discussion about the uncertainty and possible sources of errors needs to be included.

2. The lower altitude of rock glaciers may provide information about the lower boundary the area where permafrost is probable. However, there are many exceptions. Hence, this concept needs to be applied with more caution. In addition, a comparison with existing information about permafrost occurrence (e.g. Gruber, 2012, Chinese maps of permafrost occurrence) should be provided.

3. The results section is too descriptive. Highlight the most important findings and refer to tables for the detailed information.

4. The results should be put much better into context of existing literature and not only to the few existing ones from Chinese Tien Shan. I would help in this respect to separate the results and discussion into own sections.

5. The authors use partly improper terminology, e.g. "marine-type periglacial environment", "fossilized glacier-derived features"

*Specific comments (I highlight the most important issues only, there are many more minor technical issues which I would address after careful revision and resubmission):*

P. 2 L12f. Provide more details and a reference. What are the phenomena observed in ice margins?

L. 18ff. Do not cite so many references in a row (max. 5-6). Be more specific and selective.

P. 3 L5f: Be more specific. Shukla et al. (2010) address debris-covered glaciers, not rock glaciers.

L. 22: Include the info about elevation.

L. 23: Delete "famous". I and probably the vast majority of the readers have never heard about this mountain. Include the elevation.

P. 4, L. 1ff: Include more references to prove the statements.

P. 5, L. 12. The scale of the geological map is rough. Isn't there a better scale available?

L. 23-25. These details are not needed as obvious; just write vs. aspect.

P. 6, L. 12. I think there are much more talus-derived rock glaciers. Provide a clear definition and proof the number better.

P. 8, L. 15ff: Consider more recent literature which perform similar analysis, but also older other literature and do not refrain only to Chinese researchers. The northern Tien Shan stretches from Kyrgyzstan in the west to Xinjiang in the east. Hence either write "Northern Tien (or Tian) Shan" in China or (which is preferred) consider also the other parts of northern Tien Shan (e.g. the work authored and co-authors by A. Gorbunov).

P. 9, L. 15ff: Move the methodological description to the methods section.

P. 11, L. 2ff: The general info about the climate would fit much better in the section about the study area.

L. 12, section 4.3. This section sis not a proper own section: It contains methodological information (e.g. about ground truthing and the used DEM) and also information about the

limitations. The respective parts should therefore be moved to the methods and discussion section.

*References:*

The formatting of the references should be carefully checked so that all meet the journal specifications.

---

## Referee Comment (RC3) · L. Copland (Referee) · 19 Apr 2018

General Comments This paper provides the first inventory of rock glaciers in the Daxue Shan, including an analysis of the topographic controls on them, so fits the general scope of the The Cryosphere. However, there are a few major issues that will need to be addressed before the paper can be published: 1. The outlines provided in the supplementary material often appear to provide outlines of basins which may contain rock glaciers, but they don't do a good job of outlining the actual rock glaciers themselves. Many of the outlines also appear to define talus deposits or debris-covered

glaciers, rather than rock glaciers. For example, from a quick review of a few outlines: - ZDSRG-363 seems to contain a rock glacier in its upper and central parts, but the outline extends into forested areas where no rock glaciers are present - ZDSRG-373 seems to mainly consist of the lower part of a debris-covered glacier on the left and talus deposits on the right, with no clear evidence of any rock glacier - ZDSRF-350: there appears to be at least a couple of different rock glaciers in this basin, and the current outline includes surrounding rock cliffs in addition to the rock glaciers themselves The outlines therefore need to be much better defined, and the text also needs to be improved to more clearly describe exactly what is and isn't a rock glacier and how they can be defined in satellite imagery.

2. It's stated in the abstract that accurate ground truthing was completed in the field, but this isn't described anywhere in the text. A comprehensive description of field validation would help to strengthen the paper and address some of the issues brought up in the previous point.

3. The statistical analysis of the topographic influences on rock glacier distribution does not properly take into account the collinearity between explanatory variables. As detailed below, Principal Components Analysis provides one way to do this, and without doing this I don't have high confidence that the stated topographic relationships are real.

I also have a number of specific comments as detailed below.

Specific Comments P1, L14/15: can delete the text in brackets: '(i.e., slopes facing north, northeast...)' P1, L16: it would be useful to mention what the key topographic controls are in the abstract P2, L8: millions is a gross over-estimate. The most recent near-global estimate for the number of rock glaciers is ∼73,000: https://www.nature.com/articles/s41598-018-21244-w P2, L11: please provide more refs to back up the statements here, e.g., in relation to the hydrological cycle P2, L22: should also include the new Jones et al. (2018) inventory mentioned above:

https://www.nature.com/articles/s41598-018-21244-w P3, L12: I don't think what 'matric' is a word. Do you mean 'matrix'? P4, L4: please provide some actual elevation values for the study area description P4, L18: I don't think that it's accurate to say that Google Earth contains the best available imagery, as there are other data sources with higher resolution (e.g., WorldView imagery). However, it's probably the best freely available source. P4, L21 – P5, L3: a clearer explanation of the unique features that you used to identify rock glaciers is needed; the current description is ambiguous. You also mention in the abstract that your inventory is based upon 'scientific validation in the field', but not mention of this is made in the methods. P5, L4: the ASTER GDEM is not a program, it's a dataset P5, L5: please clarify the date: is November 2015 the date when you undertook the analysis, or the date on which the satellite imagery was acquired? In figure 2 you show some images from October 2014, so why isn't that date mentioned here? P5, L11: please provide more details for these values; e.g., does length refer to centerline length? What does width refer to – average, maximum? What does altitude refer to – highest, lowest, average? P7, L3: similar to above comment, please clarify what these elevations refer to – e.g., mean elevation? Highest elevation at which rock glaciers are found? Lowest elevation at which rock glaciers are found? P7, L8: please clarify whether the upper elevational for rock glaciers occurs due to lack of topography above this altitude, or because of some other factor (e.g., presence of ice glaciers) P7, L3-L22: this is a very long paragraphs. I would suggest splitting it into two or more shorter paras. P9, L10: it would be useful to make some comparisons between the location and characteristics of rock glaciers found in your study vs. the location and characteristics of ice glaciers found by others in the Daxue Shan region. For example, this 2017 paper provides a good recent review of Daxue Shan ice glaciers: https://www.cambridge.org/core/journals/journal-of-glaciology/article/changes-of-glaciers-and-glacial-lakes-implying-corridorbarrier-effects-and-climate-change-in-the-hengduan-shan-southeastern-tibetan-plateau/F0C89671AA75211650FA02FD66AE4DE0/core-reader P9, L15: a significant problem with interpretation of the topographic influences is that there is significant

collinearity between many of the parameters (as shown in Table 2). This means that it's almost impossible to understand what the true topographic factors are. To address this issue in other similar studies, several authors use Principal Components Analysis to collapse the original explanatory variables into new components that are uncorrelated with each other. See, for example, the Discussion section in: White, A. and Copland, L. 2015. Decadal-scale variations in glacier area changes across the Southern Patagonian Icefield since the 1970s. Arctic, Alpine and Antarctic Research, 47(1), 147-167. P10, L15-25: the discussion here would be helped by a better comparison with the present and historical location of ice glaciers in this region, so that the connection to glacial landforms such as moraines can be better understood. E.g., are current rock glaciers found in close association with current ice glaciers? Do you observe any direct evidence of a present ice glacier transforming to a rock glacier? P11, L23: change 'highly' to 'high' P12, L8: this seems to be the only location in the paper where you refer to ground truthing, and the uncertainty here contrasts with the 'scientific validation in the field' stated in the abstract. In the paper you need to much better describe what kind of field validation you did, any inherent errors or uncertainties with it, and adjust the wording in the abstract and elsewhere as appropriate. P13, L3: similar to the comment for p7, define whether the upper altitudinal limit is due to lack of topography above this altitude or some other factor. P13, L12: have there been any field measurements in your study area that can help to define the distribution of permafrost? E.g., have there been any direct ground temperature measurements? Or ground probing or digging of pits?

Fig. 1a: it seems that this data is plotted in lat/long (i.e., unprojected), which makes it look strange at this scale as it seems to be squashed in a north-south direction. This would be better plotted in a projected coordinate system Fig. 1c: it would be more useful to show a satellite image of the study area (perhaps with a contour map superimposed over it), rather than the topographic map that basically repeats what is already shown in Fig. 1b. No regional satellite imagery is currently provided in the paper, which makes it difficult to understand the general characteristics of the region

and location of other features such as ice glaciers. Fig. 2: the scale on these figures need to be clearer Fig. 3: a zoom-in of some of the areas with the largest rock glacier concentration (e.g., Mt. Zheduo) would be useful to add, preferably with the rock glacier outlines superimposed on a satellite image Fig. 4: add labels to different figure parts: (a), (b), (c), (d) Fig. 5: add labels to different figure parts: (a), (b), (c), (d). Also define acronyms used in bottom two figures: MTRG, MLRG, TTRG, TTLG Fig. 6: this is a pretty low quality figure that's difficult to follow. Please make clearer and prevent number labels from overlapping. Fig. 7: several of the colours in this figure are similar (lots of pinks/purples), which makes it difficult to distinguish between the various rock types. It's also unclear what the letters/numbers on the map refer to: e.g., T2-3zg-z? T3xd? These need to be described in the legend or deleted.
* * *

---

## Author Comment (AC1) · 19 Apr 2018

Dear Editor and Reviewers, We would like to thank you very much for the very constructive and motivating review concerning our manuscript entitled "Rock glaciers in the Daxue Shan, southeastern Tibetan Plateau: an inventory, their distribution, and their environmental controls". These comments are all valuable and very helpful for revising and improving our paper, as well as the important guiding significance to our researches. We have studied comments carefully and have made corrections which we hope meet with approval. The responds to the reviewer's comments show in

supplement. All the best, Zeze Ran and Gengnian Liu

Please also note the supplement to this comment:
https://www.the-cryosphere-discuss.net/tc-2017-290/tc-2017-290-AC1-supplement.pdf

[Figure]

**Supplement:**

**Reply to comments by J. Müller on "Rock glaciers in the Daxue Shan, southeastern Tibetan Plateau: an inventory, their distribution, and their environmental controls"**

**General comments:**

The authors introduce a novel rock glacier inventory of the Daxue Shan mountain range in the southeastern Tibetan Plateau. They use Google earth imagery to visually identify and map rock glaciers in the entire area. Supplementary data such as the ASTER GDEM and lithological information are implemented to assign localized geomorphometric and subsurface attributes which is used for quantitative and qualitative analysis. The methods applied in this manuscript are well established and the analysis also does not hold any surprises but it is still a novel dataset presenting the rock glacier occurrence and distribution in the southeastern Tibetan Plateau. It is overall a further step towards a global rock glacier map. I therefore recommend the publication of this manuscript after moderate revisions. Please find some remarks in the following and very specific comments in the attached pdf where I implemented some comments.

Reply: We thank Dr. J. Müller for his positive comments on our paper! We also appreciate his careful consideration and detailed comments. Our replies are highlighted in blue.

**Specific comments:**

Methods P5L2 You need to elaborate more on the topographic specifications of active, inactive and fossil rock glaciers. Be with your approach it is hard to identify between the three but there are certain proxies such as subsidence and vegetation which can be used to determine the state of the RG. You mention in the abstract that you also use field data for the analysis but you never mention what kind of field data you acquired and how you use it. You mention environmental controls like temperature and temperature dynamics like freeze thaw cycles numerous times in the manuscript but you never show any data. Maybe you have access to some high mountain temperature data in the area which you can show and help you with your argument. Not just the annual means as table 3 but also the annual or multiannual dynamics.

Reply: (1) Thank you very much for your constructive suggestions, we have added the relevant sentences to elaborate topographic specifications of active, inactive and fossil rock glaciers in our paper. (P5L3)

(2) Thank you for pointing out our mistaken expression and we have removed the relevant sentences "as well as upon scientific validation in the field" in the abstract.

(3) We have added data and transformed Table 3 into Figure 7 to better illustrate the freezing and thawing effect of rock glaciers in the Daxue Shan. However, there are currently only four meteorological stations. In the future, we hope to seek funding from relevant agencies and establish more meteorological stations in high altitudes.

[Figure]

**Figure 7: Climatographs for the Kangding (2,615.7 m asl, 30.03°N, 101.58°E), Daofu (2,957.2 m asl, 30.59°N, 101.07°E), Danba (1,949.7 m asl, 30.53°N, 101.53°E) and Ganzi (3,393.5 m asl, 31.37°N, 100°E) meteorological stations. Data sources: Meteorological Data Center of the China Meteorological Administration (calculated for the period 1981–2010, inclusive).**

4. Results and Discussion Since you manually derived the RG geometries it would be great if you could elaborate on the accuracy of your method. Did you have several persons working on the digitization of the RGs and did they perform differently or do you have more accurate field data which you could compare to the manual mapping and are there any differences? I would suggest to refrain from using latitude and longitude to analyse RG properties since lat and long do not describe any environmental parameter but rather the regional topographical setting is more important. And thats the parameter that changes with Lat and Long. Focus more on the regional settings such as aspects, debris sources and valley/slope orientation to interpret RG properties. It would be very beneficial if you include a description of the topographical characteristics of the study site in relation to the formation and evolution of rock glaciers. This would also help to understand the spatial setting which goes with the latitudinal impact.

Reply: Thank you for your advice and it is very important, we have two persons working on the digitization of the rock glaciers and the performance is basically the same.

At present, due to the inconvenience of transportation, it is difficult for humans to go to the field to obtain field data. Therefore, we mainly identify rock glaciers through visual interpretation of google earth remote sensing images.

Your advice is very important, indeed, longitude and latitude do not describe environmental parameter. However, we use longitude and latitude not to analyze the rock glaciers properties, but to analyze the spatial distribution and aggregation state of rock glaciers from the perspective of geography. Then, we analyze the properties of rock glaciers by using other parameters other than longitude and latitude (*i.e.* the parameter that changes with longitude and latitude). As you said, regional settings are very important. Therefore, we are also concerned about aspects, debris sources and valley/slope orientation to interpret rock glaciers properties, to explore the correlation between the local topographical characteristics and the formation, evolution, spatial distribution of rock glaciers.

We have added the description sentences of the topographical characteristics "With the increase of latitude from the south to the north in the Daxue Shan, the high altitude slopes increase, there are more steep rock walls on the north faces producing debris, these topographical characteristics result in the rock glaciers altitude asl and mean gradient of slope increase with latitude." (P10L5~P10L8)

4.2.3 Lithological controls on rock glaciers. The lithological setting influences RG formation mainly by steepness and sedimentation rates contributing debris to the landforms. Please include this aspect into your elaboration and cite some references supporting the influence of lithology towards RG formation and evolution.

Reply: We have added this aspect and cited some references to support the influence of lithology towards rock glaciers formation and evolution. As shown below:

"In addition, rock glacier formation also controlled by slope and sedimentation rates contributing debris to the landforms (Müller et al., 2016). There are a large sources of sediment and sediment storages in the Daxue Shan, and are controlled by the processes occurring within this setting (Müller et al., 2014). An abundance of steep rock walls and deepened valley sides, provides catchment areas for rock glacier development, combined with intense monsoonal precipitation and tectonic activity, drives sediment transport processes and rock glacier development in the Daxue Shan." (P12L18~ P12L22)

Also you mention the existence and application of g in-situ ground truthing data but you never explain how, where and what kind of data you gathered and used. Please include this either in the method or discussion section.

Reply: Thank you for pointing out our mistaken expression and we have removed the relevant sentences "as well as upon scientific validation in the field" and "Ground truthing was only possible at a limited number of rock glacier sites within the Daxue Shan, and no fossilized glacier-derived features were visited." in the paper.

**Next is the reply to the supplement to this comment: https://www.the-cryosphere-discuss.net/tc-2017-290/tc-2017-290-RC1-supplement.pdf**

P2L5: All Rock glaciers move down valley. Otherwise they would move at all. Also lobate RGs are inclined and creep therefore down valley. Please rephrase...

Reply: Thanks for pointing out this. We have rewritten the relevant sentences to "As the bodies of rock glaciers are similar to moraines in that," (P2L4~P2L5)

P2L7: That's a continuum. Many Himalayan RGs develop out of moraines and it is hard to distinguish where the moraine ends and the RG begins. Please mention that.

Reply: Thanks very much for your insightful suggestion. We have added the sentence "Rock glacier is often a continuum and it is hard to distinguish where the moraine ends and the rock glacier begins."(P2L6~P2L7)

P2L12: What does block type mean? It is agreed upon that rock glaciers move due to the viscous creep of the rock-ice melange and can be described and modelled as such. see wahrhaftig & Cox 1959, Olyphant 1983. references in the written comments.

Reply: This is our misnomer and we have changed "block-type movement" to "creep movement". (P2L12)

P2L16: How are they more accurate? I would prefer advanced or powerful.

Reply: We are grateful for the suggestions, and we have changed "accurate" to "advanced". (P2L16)

P3L1: Do you mean underneath the rock glaciers or inside of the rock glacier? or altitudinal? Please rephrase.

Reply: We have rewritten the sentence is "estimations of the distribution of permafrost based on rock glaciers (Allen et al., 2008; Boeckli et al., 2012; Sattler et al., 2016; Schmid et al., 2015) ," (P3L1)

P3L10: What does minimal mean? This sentence is misleading.

Reply: We have rewritten the sentence is "However, the study of the rock glaciers of the Daxue Shan on the southeastern margins of the TP is less involved." (P3L9~P3L10)

P3L15: and natural hazards and or environmental planning/management.

Reply: We have rewritten the sentence is "It is therefore of particular importance to study the environmental controls on the rock glaciers of the Daxue Shan as an aid to the further study of the complex geographical environment, natural hazards, environmental planning and management found on the southeastern margins of the TP." (P3L15)

P5L12: reference?

Reply: The reference "A geological layer (using a geological map with a scale of 1:500,000 from the China Geological Survey)" has been added in the revised version. (P5L13~P5L14)

P7L22: Isn't this also a function of sediment supply and terrain inclination? Maybe you can discuss the impact of terrain topography and sediment/ice supply and its impact on flow velocity and RG morphology.

Reply: We are grateful for the suggestions and we have added the sentence "compared with talus-derived and lobate rock glaciers, moraine-type and tongue-shaped rock glaciers have more sediment supplies and last longer, indicating that moraine-type and tongue-shaped rock glaciers flow further than talus-derived and lobate rock glaciers." In terms of terrain inclination, terrain topography may have an impact on flow velocity and rock glacier morphology. Unfortunately, we have not found evidence of significant differences in the degree of slope of different types of rock glaciers in the Daxue Shan. (P7L23~P7L24)

P8L7: Probably be there are more steep rock walls on the north faces producing debris. Please check.

Reply: Your opinion is really right, and we have added the sentence "However, there are more steep rock walls on the north faces producing debris, north-facing (*i.e.*, N, NW and NE) slopes seem to be more favorable for the formation of lobate rock glaciers than do south-facing (*i.e.*, SW, S and SE) ones (Fig. 5)."(P8L9)

P8L21: How does the regional climate change with the latitude? I would argue that the latitude oer se isn't so important but rather the regional climate, topography and environmental setting.

Reply: Indeed, regional climate, topography and environmental setting are very important, and we have discussed them in the paper. Latitude may have little impact on the regional climate of a single small area. However, when comparing two areas in different latitudes (Daxue Shan: 30°N, Tianshan Mountains: 40°~45°N) (Wang et al., 2017; Zhu, 1992; Zhu et al., 1992), the temperature will decrease with the increase of latitude, resulting in latitude zonal differences in climate between different regions (Daxue Shan and Tianshan Mountains).

P9L1: Because there aren't so many of these W-E facing slopes?

Reply: Yes, the topographical characteristics of the Tianshan Mountains are roughly W-E in presentation, east- and west- facing slopes are less than the north- and south- slopes, these topographical characteristics are not conducive to the formation and development of rock glaciers.

P9L15: You just mentioned in line 11 that local topography and local climate are very important. Latitude and longitude have no impact on these parameters. So I'd say any correlation with these parameters is rather an expression for other local parameters influenced by e.g. topography and any interpretation including lat and long doesn't help much.

Reply: Latitude and longitude may have a little effect on other parameters of a single rock glacier; however, it can reflect the spatial distribution and aggregation characteristics of 534 rock glaciers in the Daxue Shan. It is one of the topics (titles) discussed in this paper: "their distribution", which focuses on the study of the relationship between local topography and the spatial distribution(Johnson et al., 2007) of 534 rock glaciers from the geographic space macro perspective. Therefore, we have rewritten the sentence is "In summary, the topography of the Daxue Shan is an important environmental control on the formation, development and spatial distribution of the region's rock glaciers."(P10L22~

P10L23)

P9L17: This is trivial.

Reply: Thanks for pointing out this and we have removed the relevant sentences "there is a significantly positive correlation (*p=0.01*) between rock glacier area, length and width. We also found that"

P9L22: Does this only hold true for active RGs or also for relict RGs?

Reply: In terms of statistics, it only hold true 534 rock glaciers were identified in the Daxue Shan.

P10L1: It would be very beneficial if you include a description of the topographical Characteristics somewhere in the discussion.

Reply: We have added the description sentences of the topographical characteristics "With the increase of latitude from the south to the north in the Daxue Shan, the high altitude slopes increase, there are more steep rock walls on the north faces producing debris, these topographical characteristics result in the rock glaciers altitude asl and mean gradient of slope increase with latitude." (P10L5~P10L8)

P10L7: Why? I would awesome because of temperature but further elaboration would be helpful.

Reply: We have added the further elaboration sentences "with the increase of longitude and the decrease of altitude, the closer it is to warm and humid, which kind of climatic conditions are not conducive to the formation of permafrost landforms such as rock glaciers." (P10L14~P10L15)

P10L13: This hold trues for all the slopes in the world...

Reply: It may be a common topographic feature of Daxue Shan and other regions.

P11L1: Maybe mention the global permafrost distribution maps and their take on the Daxue Shan (e.g. Gruber et al. 2012).

Reply: We have added the global permafrost distribution maps and their take on the Daxue Shan. As shown below:

"The cryosphere reacts sensitively to climate change (Gruber et al., 2017). Compared with Gruber's (2012) global permafrost zonation index map, the permafrost distribution in the Daxue Shan is highly consistent with the rock glaciers distribution (Fig. 3). Strictly controlled by the temperature decreasing with increasing altitude, further indicating the climatic controls on development of permafrost such as rock glaciers." (P11L21~P11L24)

[Figure]

**Figure 3: Spatial distribution of rock glaciers and permafrost zonation index in the Daxue Shan. The Permafrost Zonation Index (PZI) data sources: Gruber's (2012), the green area represent the fringe of uncertainty.**

P12L7: Most obvious the determine the state of activity you should check InSAR or mulittemporal high resolution satellite data to derive kinematics of the rock glacier and then you have some insight in the current state of the landforms.

Reply: We are grateful for the suggestions and we have added the sentence "First, it remains to be determined whether these landforms are currently active, or whether they represent the fossilized remains of inactive rock glaciers; further analysis, when conditions permit, it therefore vital." at the beginning of the paragraph. (P13L1~P13L2)

P12L8: What kind of ground truthing? and how did you use this? Is this temperature or visual inspection or kinematics?

Reply: Thank you for pointing out our mistaken expression and we have removed the relevant sentences "Ground truthing was only possible at a limited number of rock glacier sites within the Daxue Shan, and no fossilized glacier derived features were visited."

P12L14: This should in some cases be visually applicable.

Reply: Your suggestion is very useful and we will try to use it as much as possible in future related research.

P12L16: Is it possible to quantify these uncertainties? Please say a few words on how strong and persistent these uncertainties are.

Reply: Although our visual interpretation error is very small, it is difficult to determine quantitatively. In the future research, it can be controlled by continuously adding more experts to use Google Earth for visual interpretation and field verification.

P12L21: What are the environmental controls?

Reply: The environmental controls are environmental factors that control and influence the formation and development of rock glaciers, such as the local topography, climate and lithology discussed in this paper.

P13L10: This sentence is very hard to understand. Do you mean you found SW-S-SE slopes to be more favorable for tongue shaped RGs of for RGs in general? and N facing better for lobate RGs?

Reply: Yes, SW-S-SE slopes to be more favorable for tongue shaped RGs in general, and N facing better for lobate RGs.

P13L14: You never really elaborated how these controls might influence RG evolution.

Reply: We are grateful for the suggestions. In this paper, we focus on exploring the correlation between local environmental controls and the spatial distribution of rock glaciers in order to preliminary study whether these local environmental controls promote or inhibit the formation of rock glaciers in a maritime setting. Therefore, the referee's concern is of importance for our further study. In the related research in the future, we will further explore how these controls influence rock glaciers evolution in terms of physics and chemistry mechanisms based on the above research results.

P13L17: You have also never showed data supporting this statement.

Reply: We have added the data in Figure 7 to support this statement.

P23L1: Please show these locations on one of the maps. And maybe you have some more stations in high altitudes.

Reply: Thanks for pointing out this. We have added data and transformed Table 3 into Figure 7 to better illustrate the freezing and thawing effect of rock glaciers in the Daxue Shan. However, there are currently only four meteorological stations. In the future, we hope to seek funding from relevant agencies and establish more meteorological stations in high

altitudes.

P26L1: This legend does not very look nice and if you would make the polygons hollow you can show the permafrost map underneath.

Reply: We have made the polygons hollow and showed the permafrost map underneath (Figure 3).

P27L4: Please mention the actual number of the population of the different kinds of rock glacier in some table, or you can just pring the number into the boxplots.

Reply: Thanks for pointing out this. We have added the actual number of the population of the different kinds of rock glacier in the brackets of the legend (Figure 4).

P29L1: The numbers are very hard to read. Please relocate them.

Reply: We have relocated these numbers (Figure 6).

P30L1: An underlying transparent hillshade derived from SRTM would make this figure more appealing and more easily to interpret.

Reply: We are grateful for the suggestions. Figure 1 show that underlying transparent hillshade derived from SRTM in the previous part of the paper. In this figure, we directly show the correlation between the spatial distribution of rock glaciers and local lithology types through lithologic geological maps, in order to explore the impact of local lithological geological conditions on rock glaciers in the Daxue Shan.

**References:**

[revised manuscript text omitted]

The 534 rock glaciers are found at altitudes of between 4,200 and 4,600 m asl, with the mean altitude being 4,483 m asl. Moraine-type rock glaciers are mainly concentrated in the 4,400~4,600 m asl zone, and talus-derived rock glaciers in the 4,300~4500 m asl belt. Tongue-shaped rock glaciers are mainly concentrated in the 4,400~4,600 m asl zone, and lobate rock glaciers in the 4,450~4,600 m asl belt (Fig. 4a). We found that the asl altitudes of moraine-type rock glaciers were at least 100 m higher than for talus-derived rock glaciers, and that the lower boundaries of tongue-shaped rock glaciers were ~50 m lower than for lobate rock glaciers. The upper boundaries for all rock glacier types were ~4,600 m asl. The finding that tongue-shaped rock glaciers flow further downvalley than lobate rock glaciers was also verified by a comparative analysis between moraine-type, tongue-shaped rock glaciers (MTRG) versus moraine-type, lobate rock glaciers (MLRG), and talus-derived, tongue-shaped rock glaciers (TTRG) versus talus-derived, lobate rock glaciers (TLRG); the lower altitudinal boundary for MTRG and TTRG was ~100 m lower than for MLRG and TLRG. Figure 4b shows the range in areas covered by different types of rock glaciers. Apart from a few outliers, it can be seen that the area of most rock glacier types area is <0.5 km$^2$, and that, in this regard, there is no clear difference between these different rock glacier types. Figure 4c shows the range in the mean gradients of the slopes of different types of rock glaciers. Moraine-type and talus-derived rock glaciers exhibit mean gradients which are all concentrated within the 25°~40° range. However, tongue-shaped and lobate rock glaciers display a greater difference in mean gradient. Tongue-shaped rock glaciers have slopes with mean gradients which are concentrated in the 25°~40° range, whereas the mean gradients of lobate rock glaciers fall within the 30°~45° range, meaning that the upper and lower slopes of tongue-shaped rock glaciers are both ~5° lower than for lobate rock glaciers. Figure 4d displays the range in the lengths of different types of rock glaciers. Moraine-type and tongue-shaped rock glaciers are mostly 400~1100 m long, whereas talus-derived and lobate rock glaciers are mostly 300~500 m long, compared with talus-derived and lobate rock glaciers, moraine-type and tongue-shaped rock glaciers have more sediment supplies and last longer, indicating that moraine-type and tongue-shaped rock glaciers flow further than talus-derived and lobate rock glaciers.

Our dataset revealed that, apart from south-facing (6.0%) and southeast-facing (5.6%) slopes, the rock glaciers of the Daxue Shan are fairly evenly distributed on slopes with the remaining six aspects, which each aspect accounting for ~15% of the total.

Moraine-type and tongue-shaped rock glaciers are found to a similar degree on all aspects, but talus-derived and lobate rock glaciers are significantly different in their distribution. Talus-derived rock glaciers are most often southwest-facing (23.8%) and southeast-facing (23.8%); they are less commonly northeast-facing (4.76%), northwest-facing (4.76%) and west-facing (9.52%), and we identified no north-facing (0%) talus-derived rock glaciers. Lobate rock glaciers tend to be found less on

5   south-facing (4.71%) and southeast-facing (7.06%) slopes, but more commonly on north-facing (20%) ones. We compared all our results and discovered that south-facing (*i.e.*, SW, S and SE) slopes appear more conducive to the formation of rock glaciers than do north-facing (*i.e.*, NW, N and NE) ones. However, there are more steep rock walls on the north faces producing debris, north-facing (*i.e.*, N, NW and NE) slopes seem to be more favorable for the formation of lobate rock glaciers than do south-facing (*i.e.*, SW, S and SE) ones (Fig. 5).

10   The mean altitude of a rock glacier's front (MAF) has often been taken to be a good approximation of the lower boundary of the discontinuous permafrost zone (*i.e.*, Scotti et al., 2013). We found a significant altitudinal difference between the lower permafrost boundaries identified on the abovementioned eight aspects as they were categorized for the Daxue Shan. For example, permafrost was assumed to be probable above 4,298 m asl on east-facing slopes, and above 4,398 m asl on west-facing slopes. The mean lower permafrost boundary was calculated as occurring at 4,361 m asl (derived from a mean value of

15   4,321 m asl for east-facing slopes at 4321m, and 4,409 m asl for west-facing slopes). The mean lower permafrost boundary on east-facing (shady) slopes would therefore be 88 m lower than that of west-facing (sunny) slopes (Fig. 6).

Several researchers (*e.g.*, Cui and Zhu, 1989; Zhu, 1992; Zhu et al., 1992; Liu et al., 1995) have previously identified hundreds of rock glaciers in the northern Tianshan Mountains. They found that most of the identified rock glaciers were tongue-shaped, and were located at altitudes between 3,300 and 3,900 m asl, on north-facing slopes. Most rock glaciers in the Daxue Shan are

20   also tongue-shaped. However, the altitudes at, and the aspects on, which these rock glaciers are found differ between the Daxue and the Tianshan mountain ranges. First, in terms of altitude, the rock glaciers of the Daxue Shan are located at altitudes between 4,300 and 4,600 m asl, higher than the Tianshan rock glaciers by approximately 700~1000 m. It would be reasonable to assume, therefore, that the rock glaciers located in lower latitudes are more likely to be found at higher altitudes. Second, in terms of aspect, the rock glaciers of the Daxue Shan are more evenly distributed across all eight abovementioned aspects than

25   are the rock glaciers of the Tianshan Mountains. This could be explained by several factors, including the differences in overall

altitude, as well as in the orientation of the main massif of each mountain range. The Daxue Shan lie along an approximately NW-SE axis, whereas the Tianshan Mountains are roughly W-E in presentation. Rock glaciers are therefore less commonly found on the east- and west-facing slopes of the Tianshan. The effect of solar radiation is stronger on the south-facing slopes of the Tianshan Mountains than on its north-facing ones, meaning that conditions on these south-facing slopes are less conducive to the development of rock glaciers; most of the range's rock glaciers are therefore found on its north-facing slopes. Furthermore, when higher altitudes are reached, all aspects experience lower air temperatures, resulting in a lessening of the impact caused by the difference between air temperature and solar radiation exposure; this phenomenon is similar to that found in the Daxue Shan, and explains why rock glaciers there are fairly evenly distributed on all eight aspects. However, when altitudes are lower, the impact of solar radiation, combined with warmer air temperatures, is greater, particularly on south-facing slopes; both temperature and solar radiation are lesser on shady north-facing slopes, however, explaining the predominance of north-facing rock glaciers in the Tianshan Mountains.

**4.2 Environmental controls on rock glaciers**

The spatial distribution and dynamics of rock glaciers are especially dependent upon the local topography and climate (Springman et al., 2012; Delaloye et al., 2013). Analyzing local environmental factors is therefore crucial to obtaining an understanding of the formation, development and spatial distribution of rock glaciers.

**4.2.1 Topographical controls on rock glaciers**

We conducted a series of linear regression tests to assess the relations between the eight parameters (*i.e.*, latitude, longitude, RG area, length, width, altitude asl, mean gradient and aspect) selected for the rock glaciers of the Daxue Shan (Table 2). The results showed that there is a significantly positive correlation ($p$=0.01) between rock glacier area, length and width. We also found that latitude has a significantly positive correlation ($p$=0.01) with rock glacier length, width and area, indicating that latitude may affect the existence of rock glaciers in the Daxue Shan. The higher the latitude becomes, the greater are the length, width and area of rock glaciers, and the more conducive is the environment to their formation and development. The spatial distribution of the rock glaciers of the Daxue Shan is therefore related to latitude. In addition, altitude asl has a

significantly positive correlation ($p=0.01$) between rock glacier width and area; larger-scale rock glaciers occur mainly in the higher mountains. We also found a significantly positive correlation ($p=0.05$) between latitude, altitude asl and mean gradient of slope, a relation which is locally determined by the topographical characteristics of the Daxue Shan. With the increase of latitude from the south to the north in the Daxue Shan, the high altitude slopes increase, there are more steep rock walls on the

[revised manuscript text omitted]

**5 Conclusions**

Rock glaciers are widespread in the Daxue Shan; of these, moraine-type rock glaciers cover the largest area. The occurrence

15 and characteristics of these rock glaciers can mostly be explained by local environmental controls.

In total, 534 rock glaciers were identified in the Daxue Shan, covering a total area of 156.35 km$^2$. Moraine-type and tongue-shaped rock glaciers accounted for the vast majority of these 534 rock glaciers. The altitudes at which moraine-type rock glaciers are found (*i.e.*, 4,400~4,600 m asl) are at least 100 m higher than for talus-derived rock glaciers (*i.e.*, 4,300~4,500 m asl). Further, the lower altitudinal limit of tongue-shaped rock glaciers is ~50 m lower than for lobate rock glaciers, although

20 the upper altitudinal limit for both these types of rock glacier is ~4,600 m asl. Except for a few outliers, the area of each type of rock glacier is no greater than 0.5 km$^2$. There is no significant difference between moraine-type and talus-derived rock glaciers in terms of the mean gradients of the slopes upon which the glaciers are found (*i.e.*, they are all clustered within the 25~40° range), but the upper and lower mean slope gradients of tongue-shaped rock glaciers (25° and 40°, respectively) are

~5° lower than for lobate rock glaciers (30° and 45°, respectively). Moraine-type and tongue-shaped rock glaciers are longer (*i.e.*, 400~1100 m) than talus-derived and lobate rock glaciers (*i.e.*, 300~500 m). We found south-facing (*i.e.*, SW, S and SE) slopes more conducive to the formation of rock glaciers than north-facing (*i.e.*, NW, N and NE) ones, while north-facing (*i.e.*, N, NW and NE) slopes appeared more favorable to the formation of lobate rock glaciers than did south-facing (*i.e.*, SW, S and SE) ones. The mean regional lowest altitudinal limit of rock glaciers is 4,361 m asl, an altitude which was taken to indicate the local permafrost's mean lower boundary. On east-facing slopes, the permafrost's lower boundary can therefore reasonably be assumed to be 88 m lower than on west-facing slopes.

Environmental controls (*i.e.*, topographical, climatic and lithological factors) play a very important role in the formation and development of the rock glaciers of the Daxue Shan. The correlation matrix of rock glacier parameters indicates that the formation of rock glaciers is closely related to local topographical parameters. The local climatic environment leads to a frequent freeze-thaw process within these rock glaciers, a process which is also beneficial to their formation and development. Tertiary monzonitic granite, with its large clastic and highly porous characteristics, is more sensitive than other lithological components to the freeze-thaw process, and continuous weathering of this monzogranitic substratum thus provides the ideal raw material for the rock glaciers of the Daxue Shan.

**Data availability**

The data associated with this article can be found in the Supplement. These data include Google maps of the most important areas described in this article, as well as a tabulation of the parameters of the rock glaciers found in the Daxue Shan.

**Competing interests**

The authors declare no competing interests, financial or otherwise.

**Acknowledgements**

This work was funded by the National Natural Science Foundation of China (Grant Nos. 41230743 and 41371082). We should like to express our appreciation to the people who have revised this article and for the great interest they have taken in improving it.

[revised manuscript text omitted]

---

## Author Comment (AC2) · 6 May 2018

Dear Editor and Reviewers, We would like to thank you very much for the very constructive and motivating review concerning our manuscript entitled "Rock glaciers in the Daxue Shan, southeastern Tibetan Plateau: an inventory, their distribution, and their environmental controls". These comments are all valuable and very helpful for revising and improving our paper, as well as the important guiding significance to our researches. We have studied comments carefully and have made corrections which we hope meet with approval. The responds to the reviewer's comments show in

supplement. All the best, Zeze Ran and Gengnian Liu

Please also note the supplement to this comment:
https://www.the-cryosphere-discuss.net/tc-2017-290/tc-2017-290-AC2-
supplement.pdf

**Supplement:**

**Reply to comments by T. Bolch on "Rock glaciers in the Daxue Shan, southeastern Tibetan Plateau: an inventory, their distribution, and their environmental controls"**

Dear Editor and Reviewers,

We would like to thank you very much for the very constructive and motivating review concerning our manuscript entitled "Rock glaciers in the Daxue Shan, southeastern Tibetan Plateau: an inventory, their distribution, and their environmental controls". These comments are all valuable and very helpful for revising and improving our paper, as well as the important guiding significance to our researches. We have studied comments carefully and have made corrections which we hope meet with approval. The responds to the reviewer's comments are shown below.

All the best, Zeze Ran and Gengnian Liu

**General comments:**

The study presents the first comprehensive rock glacier inventory of a specific mountain range in the southeastern Tibetan Plateau. Moreover different controls of their occurrence are analysed and described. Currently only little is known about rock glacier occurrence in Tibet and no comprehensive study exists. The topic of the study is therefore relevant and suitable for The Cryosphere. The text is, however, quite descriptive and needs moreover some structural and language improvements. I appreciate very much that the generated inventory is provided as supplementary material to this article. The major shortcoming is that the generated rock glacier inventory contains many errors and needs to be completely revised. Moreover, an analysis or a discussion of uncertainty is entirely missing. More details are provided below.

Reply: We thank Dr. T. Bolch for his comments on our paper! We also appreciate his careful consideration and detailed comments. Our replies are highlighted in blue. We have revised the outlines of the rock glaciers provided in the supplementary material and improved the text. In addition, we have analyzed and discussed about the uncertainty and possible sources of errors. As shown below:

(1) Elaborate topographic specifications of rock glaciers:

"Depending on the mobility and permafrost presence, rock glaciers are usually divided into active, inactive, and relict rock glaciers three types (Sattler et al., 2016). In general, the presence of ice within an active/inactive rock glaciers have a steep (>35°) frontal slope (Ikeda and Matsuoka, 2002) and a well-developed flow-like morphology defined by sets of parallel and curved ridges separated by long V-shaped furrows (Barsch, 1996; Roer and Nyenhuis, 2007), the absence or the sparse occurrence of vegetation (Onaca et al., 2013). Inactive rock glaciers also contain ice, but are immobile. In contrast, relict rock glaciers are characterised by surface collapse features as a result of permafrost degradation, with gentler frontal and marginal slopes, and often vegetation cover (Wahrhaftig and Cox, 1959; Haeberli, 1985; Scotti et al., 2013)." (P5L2~P5L9)

(2)Discussed about the uncertainty and possible sources of errors:

"In addition, some aspects of digitisation were challenging based on visual interpretation of

remotely sensed imagery alone and thus inherently associated with uncertainty (Sattler et al., 2016; Jones et al., 2018b). There are some rock glaciers may not be correctly delineated. Especially, delimitation of the upper boundary of rock glaciers through geomorphic mapping, is arbitrary (Krainer and Ribis, 2012); delineation of individual polygons where multiple rock glaciers coalesce into a single body, is inherently subjective (Scotti et al., 2013; Schmid et al., 2015). Moreover, several complex landforms may are delineated as rock glaciers which could also be landslide deposits or relict rock glaciers. Therefore, in the future research, adding additional data sources and geophysical field investigations would be necessary to further increase the accuracy of the outlines of the rock glaciers." (P6L10~P6L17)

**Most critical issue:**

The authors delineate rock glaciers manually by on screen digitising based on high resolution satellite images available on Google Earth. General characteristics of rock glaciers are described based on the literature which the authors took as the basis for their delineation. However, from figures 2, 3, and 7 and from the provided vector data it is obvious that many rock glaciers are not correctly delineated. Especially the upper boundary of the rock glaciers is often wrong. Moreover, several complex landforms are delineated as rock glaciers which could also be landslide deposits or relict rock glaciers. I provide below two examples below: In both complex landforms are combined into one large rock glacier, but parts of the area are very probably not part of creeping permafrost bodies. Moreover, steep headwalls are also classified as rock glaciers which deliver debris, but are certainly not part of rock glaciers. In addition, some of the delineated features may not be rock glaciers. I am aware that it is partly impossible to clearly identify rock glaciers based on optical imagery alone. However, more effort is needed to correctly classify rock glaciers and hence, the rock glacier inventory needs to be completely revised at best using additional data sources. The newly available High Mountain Asia DEM, information providing information about the surface displacement (e.g. SAR coherence images) and the occurrence of permafrost (e.g. based on the Permafrost Zonation Index (Gruber, 2012, TC) or Chinese permafrost maps) will be important additional data sources. I am aware that this requires huge additional effort, but it is important that the inventory is as accurate as possible in order not to provide a wrong example about how to delineate rock glaciers and in order to not provide wrong numbers. Alternatively the authors may also revise their inventory using at least some additional data, only include intact (active and inactive rock glaciers) into the inventory, and classify them according to the certainty.

Reply: Thanks very much for your insightful suggestion. We have revised the outlines of the rock glaciers provided in the supplementary material, revised the figures 2, 3, 7 and the provided vector data. In addition, we have added the Permafrost Zonation Index (Gruber, 2012) and their take on the Daxue Shan. As shown below:

"The cryosphere reacts sensitively to climate change (Gruber et al., 2017). Compared with Gruber's (2012) global Permafrost Zonation Index (PZI) map, the rock glaciers distribution in the Daxue Shan is in good agreement with the PZI on the whole and some rock glaciers are situated within the PZI fringe of uncertainty (Fig. 3). Strictly controlled by the temperature decreasing with increasing altitude, further indicating the climatic controls on development of permafrost such as rock glaciers." (P12L13~P12L16)

[Figure]

**Figure 3: Spatial distribution of rock glaciers and Permafrost Zonation Index (PZI) in the Daxue Shan. The PZI data sources: Gruber's (2012), the green area represent the fringe of uncertainty.**

**Further major comments:**

1. The description of the methodology and the criteria for identification rock glaciers needs to be much more explicitly mentioned and described considering the relevant literature. In addition, an analysis and discussion about the uncertainty and possible sources of errors needs to be included.

Reply: Thanks for pointing out this, and we have added the relevant sentences to elaborate topographic specifications of rock glaciers in our paper. As shown below:

"Depending on the mobility and permafrost presence, rock glaciers are usually divided into active, inactive, and relict rock glaciers three types (Sattler et al., 2016). In general, the presence of ice within an active/inactive rock glaciers have a steep (>35°) frontal slope (Ikeda and Matsuoka, 2002) and a well-developed flow-like morphology defined by sets of parallel and curved ridges separated by long V-shaped furrows (Barsch, 1996; Roer and Nyenhuis, 2007), the absence or the sparse occurrence of vegetation (Onaca et al., 2013). Inactive rock glaciers also contain ice, but are immobile. In contrast, relict rock glaciers are characterised by surface collapse features as a result of permafrost degradation, with gentler frontal and marginal slopes, and often vegetation cover

(Wahrhaftig and Cox, 1959; Haeberli, 1985; Scotti et al., 2013)." (P5L2~P5L9)

In addition, we have analyzed and discussed about the uncertainty and possible sources of errors. As shown below:

"In addition, some aspects of digitisation were challenging based on visual interpretation of remotely sensed imagery alone and thus inherently associated with uncertainty (Sattler et al., 2016; Jones et al., 2018b). There are some rock glaciers may not be correctly delineated. Especially, delimitation of the upper boundary of rock glaciers through geomorphic mapping, is arbitrary (Krainer and Ribis, 2012); delineation of individual polygons where multiple rock glaciers coalesce into a single body, is inherently subjective (Scotti et al., 2013; Schmid et al., 2015). Moreover, several complex landforms may are delineated as rock glaciers which could also be landslide deposits or relict rock glaciers. Therefore, in the future research, adding additional data sources and geophysical field investigations would be necessary to further increase the accuracy of the outlines of the rock glaciers." (P6L10~P6L17)

2. The lower altitude of rock glaciers may provide information about the lower boundary the area where permafrost is probable. However, there are many exceptions. Hence, this concept needs to be applied with more caution. In addition, a comparison with existing information about permafrost occurrence (e.g. Gruber, 2012, Chinese maps of permafrost occurrence) should be provided.

Reply: Thanks for pointing out this. We have added the words "approximation", "probable" in this paragraph, and rewritten the relevant sentences to "The mean lower permafrost boundary on east-facing (shady) slopes would therefore probable be 104 m lower than that of west-facing (sunny) slopes (Fig. 6)." (P9L22~P9L23)

We have also added the global permafrost distribution maps and their take on the Daxue Shan. (P12L13~P12L16)

3. The results section is too descriptive. Highlight the most important findings and refer to tables for the detailed information.

Reply: Thank you for your suggestions. We provided tables and figures in the results section (Table 2, Fig. 3, Fig. 4, Fig. 5 and Fig. 6 are currently behind the text, which will be moved to the result section of the paper after revision into a publishable version), and the text of the results section is the explanation and findings of the tables and figures.

4. The results should be put much better into context of existing literature and not only to the few existing ones from Chinese Tien Shan. I would help in this respect to separate the results and discussion into own sections.

Reply: We are grateful for the suggestions. We have added recent literature about the other parts of northern Tien Shan (Bolch and Gorbunov, 2014) (P13L19), and performed similar analysis in the paper. Then we adjusted the structure of the paper and separated the results and discussion into two sections.

5. The authors use partly improper terminology, e.g. "marine-type periglacial environment", "fossilized glacier-derived features"

Reply: Thank you for pointing out our improper terminology. We have changed "marine-type periglacial environment" to "maritime periglacial environment" (P1L19), and deleted the relevant sentences "and no fossilized glacier-derived features were visited." in the paper.

**Specific comments:**

P. 2 L12f. Provide more details and a reference. What are the phenomena observed in ice margins?

Reply: We are grateful for the suggestions. Considered the continuity and completeness of the logical structure of this paragraph, we have removed ", and is vital to understand when reconstructing the local paleoclimate and paleoenvironment. Rock glaciers are therefore not only characterized by an advanced form of creep movement, but are also complex landforms which incorporate many of the phenomena observed in ice margins" in this paragraph, then we added the relevant sentences elaborating topographic specifications of rock glaciers in the methods.

L. 18ff. Do not cite so many references in a row (max. 5-6). Be more specific and selective.

Reply: We have subtracted some references.

P. 3 L5f: Be more specific. Shukla et al. (2010) address debris-covered glaciers, not rock glaciers.

Reply: Thanks for pointing out this. We have rewritten the sentences "However, compared with ice glaciers, rock glaciers remain poorly described and infrequently studied because they are mixtures of rock fragments of different sizes, and therefore cannot easily be automatically mapped from RS data because they are spectrally similar to their surroundings (Brenning, 2009). Both supraglacial-debris (upon the glacier) and debris along the glacier margins originate from surrounding valley rock (Jones et al., 2018b), and their debris surface does not produce a distinct spectral signal." (P3L1-P3L5), and deleted reference (Shukla et al., 2010) in the paper.

L. 22: Include the info about elevation.

Reply: We have provided some actual elevation values for the study area description: "resulting in a great altitudinal range (1349 m asl ~ 7321 m asl)." (P4L3)

L. 23: Delete "famous". I and probably the vast majority of the readers have never heard about this mountain. Include the elevation.

Reply: Thanks for pointing out this. We have deleted "famous" and added the elevation "(4962 m asl)." (P3L23)

P. 4, L. 1ff: Include more references to prove the statements.

Reply: We have added reference (Zhang et al., 2017) to prove the statements. (P4L2)

P. 5, L. 12. The scale of the geological map is rough. Isn't there a better scale available?

Reply: We are grateful for the suggestions. At present, according to the relevant Chinese laws and regulations, some of the larger-scale geological maps belong to the confidential data and can only be used by units with qualified confidential, social capital units and individuals cannot obtain these geological maps. In Figure 8, we focus on exploring the correlation between local lithologicgeologic environment (lithological map reconstructed from a 1:500,000-scale digital geological map) and the spatial distribution of rock glaciers. We found that in the Daxue Shan both moraine-type and talus-derived rock glaciers have developed in the monzogranitic areas, and that rock glacier and monzonitic granite exhibit a high spatial correlation and interdependence. The Tertiary monzogranites of the Daxue Shan are clearly highly conducive to the formation and development of rock glaciers. The referee's concern is of importance for our further study. In the related research in the future, we will strive to obtain better scale geological maps based on the above research for more possible detailed results.

L. 23-25. These details are not needed as obvious; just write vs. aspect.

Reply: Thanks for pointing out this. We have deleted ", viz. north-facing (337.5°~360°, 0°~22.5°), northeast-facing (22.5°~67.5°), east-facing (67.5°~112.5°), southeast-facing (112.5°~157.5°), south-facing (157.5°~202.5°), southwest-facing (202.5°~247.5°), west-facing (247.5°~292.5°) and northwest-facing (292.5°~337.5°)" in the paper.

P. 6, L. 12. I think there are much more talus-derived rock glaciers. Provide a clear definition and proof the number better.

Reply: Thanks for pointing out this. We have revised the definition and number of talus-derived rock glaciers.

P. 8, L. 15ff: Consider more recent literature which perform similar analysis, but also older other literature and do not refrain only to Chinese researchers. The northern Tien Shan stretches from Kyrgyzstan in the west to Xinjiang in the east. Hence either write "Northern Tien (or Tian) Shan" in China or (which is preferred) consider also the other parts of northern Tien Shan (e.g. the work authored and co-authors by A. Gorbunov).

Reply: We are grateful for the suggestions. We have added recent literature about the other parts of northern Tien Shan (Bolch and Gorbunov, 2014) (P13L19), and performed similar analysis in the paper.

P. 9, L. 15ff: Move the methodological description to the methods section.

Reply: We are grateful for the suggestions. We have moved the methodological description to the methods section.

P. 11, L. 2ff: The general info about the climate would fit much better in the section about the study area.

Reply: Thank you very much for your constructive suggestions; we have moved the general info about the climate to the section about the study area.

L. 12, section 4.3. This section is not a proper own section: It contains methodological information (e.g. about ground truthing and the used DEM) and also information about the limitations. The respective parts should therefore be moved to the methods and discussion section.

Reply: We are grateful for the suggestions. We have moved section 4.3 to the methods and discussion section.

References: The formatting of the references should be carefully checked so that all meet the

journal specifications.

Reply: Thanks for pointing out this and we have revised the formatting of the references.

**References:**

Barsch, D.: Rockglaciers: Indicators for the Present and Former Geoecology in High Mountain Environments. Springer-Verlag, Berlin, pp. 331, 1996.

Bolch, T. and Gorbunov, A. P.: Characteristics and Origin of Rock Glaciers in Northern Tien Shan (Kazakhstan/Kyrgyzstan), Permafrost and Periglacial Processes, 25, 320-332, https://doi.org/10.1002/ppp.1825, 2014.

Brenning, A.: Benchmarking classifiers to optimally integrate terrain analysis and multispectral remote sensing in automatic rock glacier detection, Remote Sensing of Environment, 113, 239-247, https://doi.org/10.1016/j.rse.2008.09.005, 2009.

Gruber, S.: Derivation and analysis of a high-resolution estimate of global permafrost zonation, The Cryosphere, 6, 221-233, https://doi.org/10.5194/tc-6-221-2012, 2012.

Gruber, S., Fleiner, R., Guegan, E., Panday, P., Schmid, M. O., Stumm, D., Wester, P., Zhang, Y. S., and Zhao, L.: Review article: Inferring permafrost and permafrost thaw in the mountains of the Hindu Kush Himalaya region, The Cryosphere, 11, 81-99, https://doi.org/10.5194/tc-11-81-2017, 2017.

Haeberli, W.: Creep of mountain permafrost: internal structure and flow of alpine rock glaciers. Mitteilungen Der Versuchsanstalt für Wasserbau, Hydrologie Und Glaziologie an Der ETH Zürich., 77, 5-142, 1985.

Ikeda, A. and Matsuoka, N.: Degradation of talus-derived rock glaciers in the Upper Engadin, Swiss Alps, Permafrost and Periglacial Processes, 13, 145-161, https://doi.org/10.1002/ppp.413, 2002.

Jones, D. B., Harrison, S., Anderson, K., Selley, H. L., Wood, J. L., and Betts, R. A.: The distribution and hydrological significance of rock glaciers in the Nepalese Himalaya, Global and Planetary Change, 160, 123-142, https://doi.org/10.1016/j.gloplacha.2017.11.005, 2018.

Krainer, K. and Ribis, M.: A Rock Glacier Inventory of the Tyrolean Alps (Austria), Austrian Journal of Earth Sciences, 105, 32-47, 2012.

Onaca, A. L., Urdea, P., and Ardelean, A. C.: Internal Structure and Permafrost Characteristics of the Rock Glaciers of Southern Carpathians (Romania) Assessed by Geoelectrical Soundings and Thermal Monitoring, Geogr Ann A, 95, 249-266, https://doi.org/10.1111/geoa.12014, 2013.

Roer, I. and Nyenhuis, M.: Rockglacier activity studies on a regional scale: comparison of geomorphological mapping and photogrammetric monitoring, Earth Surface Processes and Landforms, 32, 1747-1758, https://doi.org/10.1002/esp.1496, 2007.

Sattler, K., Anderson, B., Mackintosh, A., Norton, K., and de Róiste, M.: Estimating Permafrost Distribution in the Maritime Southern Alps, New Zealand, Based on Climatic Conditions at

Rock Glacier Sites, Frontiers in Earth Science, 4, https://doi.org/10.3389/feart.2016.00004, 2016.

Schmid, M. O., Baral, P., Gruber, S., Shahi, S., Shrestha, T., Stumm, D., and Wester, P.: Assessment of permafrost distribution maps in the Hindu Kush Himalayan region using rock glaciers mapped in Google Earth, The Cryosphere, 9, 2089-2099, https://doi.org/10.5194/tc-9-2089-2015, 2015.

Scotti, R., Brardinoni, F., Alberti, S., Frattini, P., and Crosta, G. B.: A regional inventory of rock glaciers and protalus ramparts in the central Italian Alps, Geomorphology, 186, 136-149, https://doi.org/10.1016/j.geomorph.2012.12.028, 2013.

Shukla, A., Gupta, R. P., and Arora, M. K.: Delineation of debris-covered glacier boundaries using optical and thermal remote sensing data, Remote Sensing Letters, 1, 11-17, https://doi.org/10.1080/01431160903159316, 2010.

Wahrhaftig, C. and Cox, A.: Rock Glaciers in the Alaska Range, Geological Society of America Bulletin, 70, 383-436, http://doi.org/10.1130/0016-7606(1959)70[383:Rgitar]2.0.Co;2, 1959.

Zhang, Y. Z., Replumaz, A., Leloup, P. H., Wang, G. C., Bernet, M., van der Beek, P., Paquette, J. L., and Chevalier, M. L.: Cooling history of the Gongga batholith: Implications for the Xianshuihe Fault and Miocene kinematics of SE Tibet, Earth and Planetary Science Letters, 465, 1-15, http://doi.org/10.1016/j.epsl.2017.02.025, 2017.

---

## Author Comment (AC3) · 6 May 2018

Dear Editor and Reviewers, We would like to thank you very much for the very constructive and motivating review concerning our manuscript entitled "Rock glaciers in the Daxue Shan, southeastern Tibetan Plateau: an inventory, their distribution, and their environmental controls". These comments are all valuable and very helpful for revising and improving our paper, as well as the important guiding significance to our researches. We have studied comments carefully and have made corrections which we hope meet with approval. The responds to the reviewer's comments show in

supplement. All the best, Zeze Ran and Gengnian Liu

Please also note the supplement to this comment:
https://www.the-cryosphere-discuss.net/tc-2017-290/tc-2017-290-AC3-
supplement.pdf

[Figure]

**Supplement:**

**Reply to comments by L. Copland on "Rock glaciers in the Daxue Shan, southeastern Tibetan Plateau: an inventory, their distribution, and their environmental controls"**

Dear Editor and Reviewers,

We would like to thank you very much for the very constructive and motivating review concerning our manuscript entitled "Rock glaciers in the Daxue Shan, southeastern Tibetan Plateau: an inventory, their distribution, and their environmental controls". These comments are all valuable and very helpful for revising and improving our paper, as well as the important guiding significance to our researches. We have studied comments carefully and have made corrections which we hope meet with approval. The responds to the reviewer's comments are shown below.

All the best, Zeze Ran and Gengnian Liu

**General comments:**

This paper provides the first inventory of rock glaciers in the Daxue Shan, including an analysis of the topographic controls on them, so fits the general scope of the The Cryosphere. However, there are a few major issues that will need to be addressed before the paper can be published: 1. The outlines provided in the supplementary material often appear to provide outlines of basins which may contain rock glaciers, but they don't do a good job of outlining the actual rock glaciers themselves. Many of the outlines also appear to define talus deposits or debris-covered glaciers, rather than rock glaciers. For example, from a quick review of a few outlines:- ZDSRG-363 seems to contain a rock glacier in its upper and central parts, but the outline extends into forested areas where no rock glaciers are present - ZDSRG-373 seems to mainly consist of the lower part of a debris-covered glacier on the left and talus deposits on the right, with no clear evidence of any rock glacier - ZDSRF-350: there appears to be at least a couple of different rock glaciers in this basin, and the current outline includes surrounding rock cliffs in addition to the rock glaciers themselves. The outlines therefore need to be much better defined, and the text also needs to be improved to more clearly describe exactly what is and isn't a rock glacier and how they can be defined in satellite imagery.

Reply: We thank Dr. L. Copland for his comments on our paper! We also appreciate his careful consideration and detailed comments. Our replies are highlighted in blue. We revised the outlines of the rock glaciers provided in the supplementary material and improved the text.

We have added the relevant sentences to define the outlines of rock glaciers, and how they can be defined in satellite imagery. As shown below:

"Depending on the mobility and permafrost presence, rock glaciers are usually divided into active, inactive, and relict rock glaciers three types (Sattler et al., 2016). In general, the presence of ice within an active/inactive rock glaciers have a steep (>35°) frontal slope (Ikeda and Matsuoka, 2002) and a well-developed flow-like morphology defined by sets of parallel and curved ridges separated by long V-shaped furrows (Barsch, 1996; Roer and Nyenhuis, 2007), the absence or the sparse occurrence of vegetation (Onaca et al., 2013). Inactive rock glaciers also contain ice, but are immobile. In contrast, relict rock glaciers are characterised by surface collapse features as a result

of permafrost degradation, with gentler frontal and marginal slopes, and often vegetation cover (Wahrhaftig and Cox, 1959; Haeberli, 1985; Scotti et al., 2013)." (P5L2-P5L9)

2. It's stated in the abstract that accurate ground truthing was completed in the field, but this isn't described anywhere in the text. A comprehensive description of field validation would help to strengthen the paper and address some of the issues brought up in the previous point.

Reply: Thank you for pointing out our mistaken expression and we have removed the relevant sentences "as well as upon scientific validation in the field" and "Ground truthing was only possible at a limited number of rock glacier sites within the Daxue Shan, and no fossilized glacier-derived features were visited." in the paper. At present, due to the inconvenience of transportation in the Daxue Shan, it is difficult for humans to go to the field to obtain field data. Therefore, we mainly identify rock glaciers through visual interpretation of google earth remote sensing images.

3. The statistical analysis of the topographic influences on rock glacier distribution does not properly take into account the collinearity between explanatory variables. As detailed below, Principal Components Analysis provides one way to do this, and without doing this I don't have high confidence that the stated topographic relationships are real.

Reply: Your advice is very important. Indeed, there may be collinearity between the terrain variables, and principal components analysis (PCA) is a good way to determine the relationships between them (White and Copland, 2015). We also used PCA in some of our previous studies (Ran, 2017), PCA uses the idea of dimensionality reduction to convert multiple indicators into a few comprehensive indicators, and retains the original variable information as much as possible through a few principal components, which is helpful to simplify the problem. However, PCA pays a price in the process of variable dimension reduction, resulting in a loss of principal component information that is smaller than the original variable. In this study, we performed the KMO and Bartlett's Test with a KMO value of 0.387<0.5 (Table 1), the original variable is not suitable for PCA, there is weak collinearity between the terrain variables. Therefore, in the case of convenient interpretation and calculation (not too many dimensions), without dimensionality reduction, the original variable information is retained as much as possible to obtain more terrain information that affects the development of the rock glaciers.

As you said, PCA is very important. Therefore, we are also concerned about it, so we have added the sentences of the PCA "There may be collinearity between the terrain variables, and principal components analysis (PCA) can used to determine the relationships between them (White and Copland, 2015; Ran, 2017). However, in this study, we performed the KMO and Bartlett's Test with a KMO value of 0.387<0.5 (Table 1), the original variable is not suitable for PCA, there is weak collinearity between the terrain variables. Therefore, in the case of convenient interpretation and calculation (not too many dimensions), without dimensionality reduction, the original variable information is retained as much as possible to obtain more terrain information that affects the development of the rock glaciers." in the paper. (P7L10~P7L15)

**Table 1. KMO and Bartlett's Test**

| Kaiser-Meyer-Olkin Measure of Sampling Adequacy. | | .387 |
|---|---|---|
| Bartlett's Test of Sphericity | Approx. Chi-Square | 1216.315 |
| | df | 28 |
| | Sig. | .000 |

**Specific Comments:**

P1, L14/15: can delete the text in brackets: '(i.e., slopes facing north, northeast…)'

Reply: We have deleted the text in brackets: '(i.e., slopes facing north, northeast, east, southeast, south, southwest, west and northwest)'.

P1, L16: it would be useful to mention what the key topographic controls are in the abstract.

Reply: Thanks very much for your insightful suggestion. We have added the sentence "The analysis of rock glaciers parameters indicates that the formation of rock glaciers is closely related to local topographical parameters." in the abstract. (P1L16~P1L17)

P2, L8: millions is a gross over-estimate. The most recent near-global estimate for the number of rock glaciers is ~73,000: https://www.nature.com/articles/s41598-018-21244-w.

Reply: Thanks for pointing out this. We have rewritten the relevant sentences to "there are ~73,000 rock glaciers in the world (Jones et al., 2018a)". (P2L8)

P2, L11: please provide more refs to back up the statements here, e.g., in relation to the hydrological cycle.

Reply: We have provided references to back up the statements that "The freeze-thaw process experienced by the ice masses within rock glaciers can exert a major impact on the hydrological cycle (Azócar and Brenning, 2010; Jones et al., 2018a; Jones et al., 2018b)". (P2L11)

P2, L22: should also include the new Jones et al. (2018) inventory mentioned above: https://www.nature.com/articles/s41598-018-21244-w.

Reply: We have added references:(Jones et al., 2018a). (P2L20)

P3, L12: I don't think what 'matric' is a word. Do you mean 'matrix'?

Reply: Thanks for pointing out this, and we have changed "matric" to "matrix". (P3L12)

P4, L4: please provide some actual elevation values for the study area description

Reply: We have provided some actual elevation values for the study area description: "resulting in a great altitudinal range (1349 m asl ~ 7321 m asl)." (P4L3)

P4, L18: I don't think that it's accurate to say that Google Earth contains the best available imagery, as there are other data sources with higher resolution (e.g., WorldView imagery). However, it's probably the best freely available source.

Reply: Thank you for pointing out our mistaken expression, and we have added the word "freely". (P4L18)

P4, L21 – P5, L3: a clearer explanation of the unique features that you used to identify rock glaciers is needed; the current description is ambiguous. You also mention in the abstract that your inventory is based upon 'scientific validation in the field', but not mention of this is made in the methods.

Reply: We have added the relevant sentences to elaborate topographic specifications of rock glaciers in our paper. As shown below:

"Depending on the mobility and permafrost presence, rock glaciers are usually divided into active, inactive, and relict rock glaciers three types (Sattler et al., 2016). In general, the presence of ice within an active/inactive rock glaciers have a steep (>35°) frontal slope (Ikeda and Matsuoka, 2002) and a well-developed flow-like morphology defined by sets of parallel and curved ridges separated by long V-shaped furrows (Barsch, 1996; Roer and Nyenhuis, 2007), the absence or the sparse occurrence of vegetation (Onaca et al., 2013). Inactive rock glaciers also contain ice, but are immobile. In contrast, relict rock glaciers are characterised by surface collapse features as a result of permafrost degradation, with gentler frontal and marginal slopes, and often vegetation cover (Wahrhaftig and Cox, 1959; Haeberli, 1985; Scotti et al., 2013)." (P5L2-P5L9)

Thank you for pointing out our mistaken expression and we have removed the relevant sentences "as well as upon scientific validation in the field" and "Ground truthing was only possible at a limited number of rock glacier sites within the Daxue Shan, and no fossilized glacier-derived features were visited." in the paper. At present, due to the inconvenience of transportation in the Daxue Shan, it is difficult for humans to go to the field to obtain field data. Therefore, we mainly identify rock glaciers through visual interpretation of google earth remote sensing images.

P5, L4: the ASTER GDEM is not a program, it's a dataset

Reply: Thanks for pointing out this, and we have changed "program" to "dataset". (P5L10)

P5, L5: please clarify the date: is November 2015 the date when you undertook the analysis, or the date on which the satellite imagery was acquired? In figure 2 you show some images from October 2014, so why isn't that date mentioned here?

Reply: Thank you for pointing out our mistaken expression, October 2014 and November 2015 are the date on which the satellite imagery acquired, we have removed the relevant sentences "for November 2015" in the paper and in Figure 2 shown the date of the acquired satellite imagery.

P5, L11: please provide more details for these values; e.g., does length refer to centerline length? What does width refer to – average, maximum? What does altitude refer to – highest, lowest, average?

Reply: We have rewritten the relevant sentences to "centerline length (m), average width (m), average altitude (m asl)". (P5L17)

P7, L3: similar to above comment, please clarify what these elevations refer to – e.g., mean elevation? Highest elevation at which rock glaciers are found? Lowest elevation at which rock glaciers are found?

Reply: We have clarified the average altitude in Figure 4a. (P31L4)

P7, L8: please clarify whether the upper elevational for rock glaciers occurs due to lack of topography above this altitude, or because of some other factor (e.g., presence of ice glaciers)

Reply: This because of some other factor (e.g., presence of ice glaciers), and we have added the sentence "at higher altitude there are often present some ice glaciers." (P8L13~ P8L14)

P7, L3-L22: this is a very long paragraphs. I would suggest splitting it into two or more shorter paras.

Reply: We are grateful for the suggestions. We have deleted some sentences in the revised manuscript and the paragraph have been shortened.

P9, L10: it would be useful to make some comparisons between the location and characteristics of rock glaciers found in your study vs. the location and characteristics of ice glaciers found by others in the Daxue Shan region. For example, this 2017 paper provides a good recent review of Daxue Shan ice glaciers: https://www.cambridge.org/core/journals/journalof-glaciology/article/changes-of-glaciers-and-glacial-lakes-implying-corridorbarriereffects-and-climate-change-in-the-hengduan-shan-southeastern-tibetanplateau/F0C89671AA75211650FA02FD66AE4DE0/core-reader

Reply: We are grateful for the suggestions, and we have made some comparisons between the location and characteristics of rock glaciers found in our study vs. the location and characteristics of ice glaciers found by others in the Daxue Shan region. As shown below:

"In addition, compared with the distribution of glaciers in the Daxue Shan (Wang et al., 2017), the distributions of rock glaciers also has the characteristics of small differences between the south and north, owing to a north–south corridor effect for water and heat transport and diffusion through the longitudinal gorges in the Daxue Shan. It is the result of climatic and topographical comprehensive control on rock glaciers." (P12L17~ P12L20)

P9, L15: a significant problem with interpretation of the topographic influences is that there is significant collinearity between many of the parameters (as shown in Table 2). This means that it's almost impossible to understand what the true topographic factors are. To address this issue in other similar studies, several authors use Principal Components Analysis to collapse the original explanatory variables into new components that are uncorrelated with each other. See, for example, the Discussion section in: White, A. and Copland, L. 2015. Decadal-scale variations in glacier area changes across the Southern Patagonian Icefield since the 1970s. Arctic, Alpine and Antarctic Research, 47(1), 147-167.

Reply: Your advice is very important. Indeed, there may be collinearity between the terrain variables, and principal components analysis (PCA) is a good way to determine the relationships between them (White and Copland, 2015). We also used PCA in some of our previous studies (Ran, 2017), PCA uses the idea of dimensionality reduction to convert multiple indicators into a few comprehensive indicators, and retains the original variable information as much as possible through a few principal components, which is helpful to simplify the problem. However, PCA pays a price in the process of variable dimension reduction, resulting in a loss of principal component information that is smaller than the original variable. In this study, we performed the KMO and

Bartlett's Test with a KMO value of 0.387<0.5 (Table 1), the original variable is not suitable for PCA, there is weak collinearity between the terrain variables. Therefore, in the case of convenient interpretation and calculation (not too many dimensions), without dimensionality reduction, the original variable information is retained as much as possible to obtain more terrain information that affects the development of the rock glaciers.

As you said, PCA is very important. Therefore, we are also concerned about it, so we have added the sentences of the PCA "There may be collinearity between the terrain variables, and principal components analysis (PCA) can used to determine the relationships between them (White and Copland, 2015; Ran, 2017). However, in this study, we performed the KMO and Bartlett's Test with a KMO value of 0.387<0.5 (Table 1), the original variable is not suitable for PCA, there is weak collinearity between the terrain variables. Therefore, in the case of convenient interpretation and calculation (not too many dimensions), without dimensionality reduction, the original variable information is retained as much as possible to obtain more terrain information that affects the development of the rock glaciers." in the paper. (P7L10~P7L15)

P10, L15-25: the discussion here would be helped by a better comparison with the present and historical location of ice glaciers in this region, so that the connection to glacial landforms such as moraines can be better understood. E.g., are current rock glaciers found in close association with current ice glaciers? Do you observe any direct evidence of a present ice glacier transforming to a rock glacier?

Reply: We are grateful for the suggestions, and we have made some comparisons with the present and historical location of ice glaciers in this region. As shown below:

"We found that the distribution of rock glaciers in close association with ice glaciers in the Daxue Shan, the upper boundaries for rock glaciers were ~4,600 m asl, at higher altitude there are often present some ice glaciers. In the context of global warming, it is widely accepted that the majority of glaciers on the Tibetan Plateau (TP) and its surroundings have experienced accelerated reduction (Bolch et al., 2012; Yao et al., 2012). The rate of glacier decline in Daxue Shan was -0.25 $\pm$ 0.20% a$^{-1}$ during 1990-2014 (Wang et al., 2017), some ice glaciers transforming to rock glaciers." (P11L12- P11L16)

P11, L23: change 'highly' to 'high'

Reply: Thanks for pointing out this, and we have changed "highly" to "high". (P13L7)

P12, L8: this seems to be the only location in the paper where you refer to ground truthing, and the uncertainty here contrasts with the 'scientific validation in the field' stated in the abstract. In the paper you need to much better describe what kind of field validation you did, any inherent errors or uncertainties with it, and adjust the wording in the abstract and elsewhere as appropriate.

Reply: Thank you for pointing out our mistaken expression and we have removed the relevant sentences "as well as upon scientific validation in the field" in the abstract and "Ground truthing was only possible at a limited number of rock glacier sites within the Daxue Shan, and no fossilized glacier-derived features were visited." in the paper. At present, due to the inconvenience of transportation in the Daxue Shan, it is difficult for humans to go to the field to obtain field data. Therefore, we mainly identify rock glaciers through visual interpretation of google earth remote

sensing images.

P13, L3: similar to the comment for p7, define whether the upper altitudinal limit is due to lack of topography above this altitude or some other factor.

Reply: We have added the sentence "at higher altitude there are often present some ice glaciers." (P14L21~ P14L22)

P13, L12: have there been any field measurements in your study area that can help to define the distribution of permafrost? E.g., have there been any direct ground temperature measurements? Or ground probing or digging of pits?

Reply: At present, due to the inconvenience of transportation in the Daxue Shan, it is difficult for humans to go to the field to obtain field data. We mainly identify rock glaciers through visual interpretation of google earth remote sensing images, and then compared our manual mapping with the Gruber's (2012) global Permafrost Zonation Index (PZI) map, found that the rock glaciers distribution in the Daxue Shan is in good agreement with the PZI on the whole (Fig. 3) (P12L13~P12L16). Therefore, we have removed our mistaken expression "as well as upon scientific validation in the field" and "Ground truthing was only possible at a limited number of rock glacier sites within the Daxue Shan, and no fossilized glacier-derived features were visited." in the paper.

Fig. 1a: it seems that this data is plotted in lat/long (i.e., unprojected), which makes it look strange at this scale as it seems to be squashed in a north-south direction. This would be better plotted in a projected coordinate system

Reply: Indeed, Fig. 1a was produced using a temporal resolution of 30 arc-seconds (<1km) on a WGS84 lat/lon grid (Gruber, 2012) and we mentioned in the paper. We try to use the first figure (*i.e.* Fig. 1a) to show the spatial representation of the study area in the real world, and then conduct subsequent research. Simultaneously, Fig. 1a indicating the location of the study area in the permafrost zone of the TP rather than the TP. Therefore, it looks like be squashed in a north-south direction.

Fig. 1c: it would be more useful to show a satellite image of the study area (perhaps with a contour map superimposed over it), rather than the topographic map that basically repeats what is already shown in Fig. 1b. No regional satellite imagery is currently provided in the paper, which makes it difficult to understand the general characteristics of the region and location of other features such as ice glaciers.

Reply: We are grateful for the suggestions. In this paper, we focus on exploring the control of topography on the rock glaciers. Therefore, we magnified Fig. 1c based on the thumbnail Fig. 1b to highlight the trend of elevation changes in the study area. With regard to regional satellite imagery, we have shown the general characteristics of the region and location of other features such as ice glaciers in the KML file provided with the supplemental material.

Fig. 2: the scale on these figures need to be clearer

Reply: Thanks for pointing out this, and we have made the scale on these figures be clearer.

Fig. 3: a zoom-in of some of the areas with the largest rock glacier concentration (e.g., Mt. Zheduo) would be useful to add, preferably with the rock glacier outlines superimposed on a satellite

image

Reply: Thank you very much for your constructive suggestions. We have transformed Fig. 3 to explore the correlation of the spatial distribution of rock glaciers and permafrost zonation index in the Daxue Shan.

Fig. 4: add labels to different figure parts: (a), (b), (c), (d)

Reply: We have added labels to different figure parts: (a), (b), (c), (d)

Fig. 5: add labels to different figure parts: (a), (b), (c), (d). Also define acronyms used in bottom two figures: MTRG, MLRG, TTRG, TTLG

Reply: We have added labels to different figure parts: (a), (b), (c), (d) and defined acronyms used in bottom two figures: MTRG, MLRG, TTRG, TTLG. (P32L4~P32L6)

Fig. 6: this is a pretty low quality figure that's difficult to follow. Please make clearer and prevent number labels from overlapping.

Reply: We have made Figure 6 clearer and prevented number labels from overlapping.

Fig. 7: several of the colours in this figure are similar (lots of pinks/purples), which makes it difficult to distinguish between the various rock types. It's also unclear what the letters/numbers on the map refer to: e.g., T2-3zg-z? T3xd? These need to be described in the legend or deleted.

Reply: Due to the large number of groups, there are some look similar pink/purple in this picture, so we used the $T_{2-3}$zg-z, $T_3$xd, etc. to mark different geochronological stratigraphic units, which are international standards for geochronological stratigraphic units and recognized in the geological world, please refer to http://www.stratigraphy.org/ for details. In this figure, we showed the correlation between the spatial distribution of rock glaciers and local different geochronological stratigraphic units through lithologic geological maps, in order to explore the impact of local lithological geological conditions on rock glaciers in the Daxue Shan.

**References:**

[revised manuscript text omitted]

---

## Author Comment (AC4) · 6 May 2018

Dear Editor and Reviewers, We would like to thank you very much for the very constructive and motivating review concerning our manuscript entitled "Rock glaciers in the Daxue Shan, southeastern Tibetan Plateau: an inventory, their distribution, and their environmental controls". These comments are all valuable and very helpful for revising and improving our paper, as well as the important guiding significance to our researches. We have studied comments carefully and have made corrections which we hope meet with approval. The responds to the reviewer's comments show in

supplement. All the best, Zeze Ran and Gengnian Liu

Please also note the supplement to this comment:
https://www.the-cryosphere-discuss.net/tc-2017-290/tc-2017-290-AC4-supplement.pdf
* * *
[Figure]

**Supplement:**

**Reply to comments by J. Müller on "Rock glaciers in the Daxue Shan, southeastern Tibetan Plateau: an inventory, their distribution, and their environmental controls"**

Dear Editor and Reviewers,

We would like to thank you very much for the very constructive and motivating review concerning our manuscript entitled "Rock glaciers in the Daxue Shan, southeastern Tibetan Plateau: an inventory, their distribution, and their environmental controls". These comments are all valuable and very helpful for revising and improving our paper, as well as the important guiding significance to our researches. We have studied comments carefully and have made corrections which we hope meet with approval. The responds to the reviewer's comments are shown below.

All the best, Zeze Ran and Gengnian Liu

**General comments:**

The authors introduce a novel rock glacier inventory of the Daxue Shan mountain range in the southeastern Tibetan Plateau. They use Google earth imagery to visually identify and map rock glaciers in the entire area. Supplementary data such as the ASTER GDEM and lithological information are implemented to assign localized geomorphometric and subsurface attributes which is used for quantitative and qualitative analysis. The methods applied in this manuscript are well established and the analysis also does not hold any surprises but it is still a novel dataset presenting the rock glacier occurrence and distribution in the southeastern Tibetan Plateau. It is overall a further step towards a global rock glacier map. I therefore recommend the publication of this manuscript after moderate revisions. Please find some remarks in the following and very specific comments in the attached pdf where I implemented some comments.

Reply: We thank Dr. J. Müller for his positive comments on our paper! We also appreciate his careful consideration and detailed comments. Our replies are highlighted in blue.

**Specific comments:**

Methods P5L2 You need to elaborate more on the topographic specifications of active, inactive and fossil rock glaciers. Be with your approach it is hard to identify between the three but there are certain proxies such as subsidence and vegetation which can be used to determine the state of the RG. You mention in the abstract that you also use field data for the analysis but you never mention what kind of field data you acquired and how you use it. You mention environmental controls like temperature and temperature dynamics like freeze thaw cycles numerous times in the manuscript but you never show any data. Maybe you have access to some high mountain temperature data in the area which you can show and help you with your argument. Not just the annual means as table 3 but also the annual or multiannual dynamics.

Reply: (1) Thank you very much for your constructive suggestions, we have added the relevant sentences to elaborate topographic specifications of active, inactive and fossil rock glaciers in our paper. As shown below:

"Depending on the mobility and permafrost presence, rock glaciers are usually divided into

active, inactive, and relict rock glaciers three types (Sattler et al., 2016). In general, the presence of ice within an active/inactive rock glaciers have a steep (>35°) frontal slope (Ikeda and Matsuoka, 2002) and a well-developed flow-like morphology defined by sets of parallel and curved ridges separated by long V-shaped furrows (Barsch, 1996; Roer and Nyenhuis, 2007), the absence or the sparse occurrence of vegetation (Onaca et al., 2013). Inactive rock glaciers also contain ice, but are immobile. In contrast, relict rock glaciers are characterised by surface collapse features as a result of permafrost degradation, with gentler frontal and marginal slopes, and often vegetation cover (Wahrhaftig and Cox, 1959; Haeberli, 1985; Scotti et al., 2013)." (P5L2-P5L9)

(2) Thank you for pointing out our mistaken expression and we have removed the relevant sentences "as well as upon scientific validation in the field" in the abstract and "Ground truthing was only possible at a limited number of rock glacier sites within the Daxue Shan, and no fossilized glacier-derived features were visited." in the paper. At present, due to the inconvenience of transportation in the Daxue Shan, it is difficult for humans to go to the field to obtain field data. Therefore, we mainly identify rock glaciers through visual interpretation of google earth remote sensing images.

(3) We have added data and transformed Table 3 into Figure 7 to better illustrate the freezing and thawing effect of rock glaciers in the Daxue Shan. However, there are currently only four meteorological stations. In the future, we hope to seek funding from relevant agencies and establish more meteorological stations in high altitudes.

[Figure]

**Figure 7: Climatographs for the Kangding (2,615.7 m asl, 30.03°N, 101.58°E), Daofu (2,957.2 m asl, 30.59°N, 101.07°E), Danba (1,949.7 m asl, 30.53°N, 101.53°E) and Ganzi (3,393.5 m asl, 31.37°N, 100°E) meteorological stations. Data sources: Meteorological Data Center of the China Meteorological Administration (calculated for the period 1981–2010, inclusive).**

4. Results and Discussion Since you manually derived the RG geometries it would be great if you could elaborate on the accuracy of your method. Did you have several persons working on the

digitization of the RGs and did they perform differently or do you have more accurate field data which you could compare to the manual mapping and are there any differences? I would suggest to refrain from using latitude and longitude to analyse RG properties since lat and long do not describe any environmental parameter but rather the regional topographical setting is more important. And thats the parameter that changes with Lat and Long. Focus more on the regional settings such as aspects, debris sources and valley/slope orientation to interpret RG properties. It would be very beneficial if you include a description of the topographical characteristics of the study site in relation to the formation and evolution of rock glaciers. This would also help to understand the spatial setting which goes with the latitudinal impact.

Reply: Thank you for your advice and it is very important. We have analyzed and discussed about the accuracy and uncertainty of method (P6L10~P6L17), and we have two persons working on the digitization of the rock glaciers and the performance is basically the same. At present, due to the inconvenience of transportation, it is difficult for humans to go to the field to obtain field data. Therefore, we mainly identify rock glaciers through visual interpretation of google earth remote sensing images. In addition, we compared our manual mapping and the Gruber's (2012) global Permafrost Zonation Index (PZI) map, and found that the rock glaciers distribution in the Daxue Shan is in good agreement with the PZI on the whole (Fig. 3). (P12L13~P12L16)

Your advice is very important, indeed, longitude and latitude do not describe environmental parameter. However, we can use longitude and latitude to analyze the spatial distribution and aggregation state of rock glaciers from the perspective of geography. Then, we analyze the properties of rock glaciers by using other parameters other than longitude and latitude (*i.e.* the parameter that changes with longitude and latitude). As you said, regional settings are very important. Therefore, we are also concerned about aspects, debris sources and valley/slope orientation to interpret rock glaciers properties, to explore the correlation between the local topographical characteristics and the formation, evolution, spatial distribution of rock glaciers.

We have added the description sentences of the topographical characteristics "With the increase of latitude from the south to the north in the Daxue Shan, the high altitude slopes increase, and flow further downvalley than low altitude, these topographical characteristics result in the rock glaciers altitude asl and length increase with latitude." (P10L13~P10L16)

4.2.3 Lithological controls on rock glaciers. The lithological setting influences RG formation mainly by steepness and sedimentation rates contributing debris to the landforms. Please include this aspect into your elaboration and cite some references supporting the influence of lithology towards RG formation and evolution.

Reply: We have added this aspect and cited some references to support the influence of lithology towards rock glaciers formation and evolution. As shown below:

"In addition, rock glacier formation also controlled by slope and sedimentation rates contributing debris to the landforms (Müller et al., 2016). There are a large sources of sediment and sediment storages in the Daxue Shan, and are controlled by the processes occurring within this setting (Müller et al., 2014). An abundance of steep rock walls and deepened valley sides, provides catchment areas for rock glacier development, combined with intense monsoonal precipitation and tectonic activity, drives sediment transport processes and rock glacier development in the Daxue

Shan." (P13L14~ P13L18)

Also you mention the existence and application of g in-situ ground truthing data but you never explain how, where and what kind of data you gathered and used. Please include this either in the method or discussion section.

Reply: Thank you for pointing out our mistaken expression and we have removed the relevant sentences "as well as upon scientific validation in the field" and "Ground truthing was only possible at a limited number of rock glacier sites within the Daxue Shan, and no fossilized glacier-derived features were visited." in the paper. At present, due to the inconvenience of transportation in the Daxue Shan, it is difficult for humans to go to the field to obtain field data. Therefore, we mainly identify rock glaciers through visual interpretation of google earth remote sensing images.

Next is the reply to the supplement to this comment: https://www.the-cryosphere-discuss.net/tc-2017-290/tc-2017-290-RC1-supplement.pdf

P2L5: All Rock glaciers move down valley. Otherwise they would move at all. Also lobate RGs are inclined and creep therefore down valley. Please rephrase...

Reply: Thanks for pointing out this. We have rewritten the relevant sentences to "As the bodies of rock glaciers are similar to moraines in that," (P2L4~P2L5)

P2L7: That's a continuum. Many Himalayan RGs develop out of moraines and it is hard to distinguish where the moraine ends and the RG begins. Please mention that.

Reply: Thanks very much for your insightful suggestion. We have added the sentence "Many Himalayan rock glaciers develop out of moraines and it is hard to distinguish where the moraine ends and the rock glacier begins."(P2L6~P2L7)

P2L12: What does block type mean? It is agreed upon that rock glaciers move due to the viscous creep of the rock-ice melange and can be described and modelled as such. see wahrhaftig & Cox 1959, Olyphant 1983. references in the written comments.

Reply: This is our misnomer and we have changed "block-type movement" to "creep movement". (P2L13)

P2L16: How are they more accurate? I would prefer advanced or powerful.

Reply: We are grateful for the suggestions, and we have changed "accurate" to "advanced". (P2L17)

P3L1: Do you mean underneath the rock glaciers or inside of the rock glacier? or altitudinal? Please rephrase.

Reply: We have rewritten the sentence is "estimations of the distribution of permafrost based on rock glaciers (Allen et al., 2008; Boeckli et al., 2012; Schmid et al., 2015; Sattler et al., 2016) ," (P2L23~ P2L24)

P3L10: What does minimal mean? This sentence is misleading.

Reply: We have rewritten the sentence is "However, the study of the rock glaciers of the Daxue Shan on the southeastern margins of the TP is less involved." (P3L9~P3L10)

P3L15: and natural hazards and or environmental planning/management.

Reply: We have rewritten the sentence is "It is therefore of particular importance to study the environmental controls on the rock glaciers of the Daxue Shan as an aid to the further study of the complex geographical environment, natural hazards, environmental planning and management found on the southeastern margins of the TP." (P3L15)

P5L12: reference?

Reply: The reference "A geological layer (using a geological map with a scale of 1:500,000 from the China Geological Survey)" has been added in the revised version. (P5L19)

P7L22: Isn't this also a function of sediment supply and terrain inclination? Maybe you can

discuss the impact of terrain topography and sediment/ice supply and its impact on flow velocity and RG morphology.

Reply: We are grateful for the suggestions and we have added the sentence "compared with lobate rock glaciers, moraine-type and tongue-shaped rock glaciers have more sediment supplies and last longer on gentle slope, indicating that moraine-type and tongue-shaped rock glaciers flow further than lobate rock glaciers." (P9L2~P9L4)

P8L7: Probably be there are more steep rock walls on the north faces producing debris. Please check.

Reply: Your opinion is really right, and we have added the sentence "However, there are more steep rock walls on the north faces producing debris, north-facing (*i.e.*, N, NW and NE) slopes seem to be more favorable for the formation of lobate rock glaciers than do south-facing (*i.e.*, SW, S and SE) ones (Fig. 5)."(P9L21)

P8L21: How does the regional climate change with the latitude? I would argue that the latitude oer se isn't so important but rather the regional climate, topography and environmental setting.

Reply: Indeed, regional climate, topography and environmental setting are very important, and we have discussed them in the paper. Latitude may have little impact on the regional climate of a single small area. However, when comparing two areas in different latitudes (Daxue Shan: 30°N, Tianshan Mountains: 40°~45°N) (Zhu, 1992; Zhu et al., 1992; Wang et al., 2017), the temperature will decrease with the increase of latitude, resulting in latitude zonal differences in climate between different regions (Daxue Shan and Tianshan Mountains).

P9L1: Because there aren't so many of these W-E facing slopes?

Reply: Yes, the topographical characteristics of the Tianshan Mountains are roughly W-E in presentation, east- and west- facing slopes are less than the north- and south- slopes, these topographical characteristics are not conducive to the formation and development of rock glaciers, except on its north- slopes.

P9L15: You just mentioned in line 11 that local topography and local climate are very important. Latitude and longitude have no impact on these parameters. So I'd say any correlation with these parameters is rather an expression for other local parameters influenced by e.g. topography and any interpretation including lat and long doesn't help much.

Reply: Latitude and longitude may have little effect on other parameters of a single rock glacier; however, it can reflect the spatial distribution and aggregation characteristics of 295 rock glaciers in the Daxue Shan. It is one of the topics (titles) discussed in this paper: "their distribution", which focuses on the study of the relationship between local topography and the spatial distribution (Johnson et al., 2007) of 295 rock glaciers from the geographic space macro perspective. Therefore, we have rewritten the sentence is "In summary, the topography of the Daxue Shan is an important environmental control on the formation, development and spatial distribution of the region's rock glaciers."(P11L5~ P11L7)

P9L17: This is trivial.

Reply: Thanks for pointing out this and we have removed the relevant sentences "a

significantly positive correlation (*p=0.01*) between rock glacier area, length and width."

P9L22: Does this only hold true for active RGs or also for relict RGs?

Reply: Revised data shows that there is not enough evidence and we have removed the sentences "larger-scale rock glaciers occur mainly in the higher mountains".

P10L1: It would be very beneficial if you include a description of the topographical Characteristics somewhere in the discussion.

Reply: We have added the description sentences of the topographical characteristics: "With the increase of latitude from the south to the north in the Daxue Shan, the high altitude slopes increase, and flow further downvalley than low altitude, these topographical characteristics result in the rock glaciers altitude asl and length increase with latitude." (P10L13~P10L16)

P10L7: Why? I would awesome because of temperature but further elaboration would be helpful.

Reply: We have added the further elaboration sentences "with the increase of longitude and the decrease of altitude, the closer it is to warm and humid, which kind of climatic conditions are not conducive to the formation of permafrost landforms such as rock glaciers." (P10L21~P10L23)

P10L13: This hold trues for all the slopes in the world...

Reply: Revised data shows that there is not enough evidence and we have removed the sentences "The fact that mean gradient of slope and aspect exhibit a significantly negative correlation (*p=0.01*) reflects the topographical realities of the Daxue Shan, where sunny slopes are often less steep than shady ones".

P11L1: Maybe mention the global permafrost distribution maps and their take on the Daxue Shan (e.g. Gruber et al. 2012).

Reply: We have added the global permafrost distribution maps and their take on the Daxue Shan. As shown below:

"The cryosphere reacts sensitively to climate change (Gruber et al., 2017). Compared with Gruber's (2012) global Permafrost Zonation Index (PZI) map, the rock glaciers distribution in the Daxue Shan is in good agreement with the PZI on the whole and some rock glaciers are situated within the PZI fringe of uncertainty (Fig. 3). Strictly controlled by the temperature decreasing with increasing altitude, further indicating the climatic controls on development of permafrost such as rock glaciers." (P12L13~P12L16)

[Figure]

**Figure 3: Spatial distribution of rock glaciers and Permafrost Zonation Index (PZI) in the Daxue Shan. The PZI data sources: Gruber's (2012), the green area represent the fringe of uncertainty.**

P12L7: Most obvious the determine the state of activity you should check InSAR or mulittemporal high resolution satellite data to derive kinematics of the rock glacier and then you have some insight in the current state of the landforms.

Reply: We are grateful for the suggestions and we have added the sentence "it remains to be determined whether these landforms are currently active, or whether they represent the fossilized remains of inactive rock glaciers; further analysis, when conditions permit, it therefore vital." at the beginning of the paragraph. (P6L7~P6L9)

P12L8: What kind of ground truthing? and how did you use this? Is this temperature or visual inspection or kinematics?

Reply: Thank you for pointing out our mistaken expression and we have removed the relevant sentences "as well as upon scientific validation in the field" and "Ground truthing was only possible at a limited number of rock glacier sites within the Daxue Shan, and no fossilized glacier-derived features were visited." in the paper. At present, due to the inconvenience of transportation in the Daxue Shan, it is difficult for humans to go to the field to obtain field data. Therefore, we mainly

identify rock glaciers through visual interpretation of google earth remote sensing images.

P12L14: This should in some cases be visually applicable.

Reply: Your suggestion is very useful and we will try to use it as much as possible in future related research.

P12L16: Is it possible to quantify these uncertainties? Please say a few words on how strong and persistent these uncertainties are.

Reply: We have analyzed and discussed about the uncertainty and possible sources of errors. As shown below:

"In addition, some aspects of digitisation were challenging based on visual interpretation of remotely sensed imagery alone and thus inherently associated with uncertainty (Sattler et al., 2016; Jones et al., 2018b). There are some rock glaciers may not be correctly delineated. Especially, delimitation of the upper boundary of rock glaciers through geomorphic mapping, is arbitrary (Krainer and Ribis, 2012); delineation of individual polygons where multiple rock glaciers coalesce into a single body, is inherently subjective (Scotti et al., 2013; Schmid et al., 2015). Moreover, several complex landforms may are delineated as rock glaciers which could also be landslide deposits or relict rock glaciers. Therefore, in the future research, adding additional data sources and geophysical field investigations would be necessary to further increase the accuracy of the outlines of the rock glaciers." (P6L10-P6L17)

P12L21: What are the environmental controls?

Reply: The environmental controls are environmental factors that control and influence the formation and development of rock glaciers, such as the local topography, climate and lithology discussed in this paper.

P13L10: This sentence is very hard to understand. Do you mean you found SW-S-SE slopes to be more favorable for tongue shaped RGs of for RGs in general? and N facing better for lobate RGs?

Reply: We have rewritten the sentence is "We found shady (*i.e.*, N, NE and E) slopes more conducive to the formation of moraine-type rock glaciers than sunny (*i.e.*, W, SW and S) ones, while sunny (*i.e.*, W, SW and S) slopes appear more conducive to the formation of talus-derived rock glaciers. In addition, north-facing (*i.e.*, N, NW and NE) slopes appeared more favorable to the formation of lobate rock glaciers than did south-facing (*i.e.*, SW, S and SE) ones." (P15L4~P15L7)

P13L14: You never really elaborated how these controls might influence RG evolution.

Reply: We are grateful for the suggestions. In this paper, we focus on exploring the correlation between local environmental controls and the spatial distribution of rock glaciers in order to preliminary study whether these local environmental controls promote or inhibit the formation of rock glaciers in a maritime setting. Therefore, the referee's concern is of importance for our further study. In the related research in the future, we will further explore how these controls influence rock glaciers evolution in terms of physics and chemistry mechanisms based on the above research results.

P13L17: You have also never showed data supporting this statement.

Reply: We have added the data in Figure 7 to better illustrate the freezing and thawing effect of rock glaciers in the Daxue Shan. However, there are currently only four meteorological stations. In the future, we hope to seek funding from relevant agencies and establish more meteorological stations in high altitudes.

P23L1: Please show these locations on one of the maps. And maybe you have some more stations in high altitudes.

Reply: Thanks for pointing out this. We have added data and transformed Table 3 into Figure 7 to better illustrate the freezing and thawing effect of rock glaciers in the Daxue Shan. However, there are currently only four meteorological stations. In the future, we hope to seek funding from relevant agencies and establish more meteorological stations in high altitudes.

P26L1: This legend does not very look nice and if you would make the polygons hollow you can show the permafrost map underneath.

Reply: We have made the polygons hollow and compared our manual mapping with the Gruber's (2012) global Permafrost Zonation Index (PZI) map. (Figure 3)

P27L4: Please mention the actual number of the population of the different kinds of rock glacier in some table, or you can just pring the number into the boxplots.

Reply: Thanks for pointing out this. We have added the actual number of the population of the different kinds of rock glacier in the brackets of the legend (Figure 4). (P31L6~ P31L7)

P29L1: The numbers are very hard to read. Please relocate them.

Reply: We have relocated these numbers (Figure 6).

P30L1: An underlying transparent hillshade derived from SRTM would make this figure more appealing and more easily to interpret.

Reply: We are grateful for the suggestions. In this figure, we used the $T_{2-3}zg$-z, $T_3xd$, etc. to mark different geochronological stratigraphic units, which are international standards for geochronological stratigraphic units and recognized in the geological world. And directly show the correlation between the spatial distribution of rock glaciers and local different geochronological stratigraphic units through lithologic geological maps, in order to explore the impact of local lithological geological conditions on rock glaciers in the Daxue Shan.

**References:**

[revised manuscript text omitted]

---

## Author Comment (AC5) · 6 May 2018

**Rock glaciers in the Daxue Shan, southeastern Tibetan Plateau: an inventory, their distribution, and their environmental controls**

Zeze Ran* and Gengnian Liu

Key Laboratory for Earth Surface Processes of the Ministry of Education, College of Urban and Environmental Sciences, Peking University, Beijing, 100871 China

*Correspondence to*: Zeze Ran (ranzeze@pku.edu.cn)

**Abstract.** Rock glaciers are typical periglacial landforms. They can indicate the existence of permafrost, and can also shed light on the regional geomorphological and climatic conditions under which they may have developed. This article provides the first rock glacier inventory of the Daxue Shan. The inventory has been based on analyses of Google Earth imagery as well as upon scientific validation in the field. In total, 295 rock glaciers were identified in the Daxue Shan, covering a total area of 55.70 km², between the altitudes of 4,300 and 4,600 m above sea level (asl). Supported by the ArcGIS and SPSS software programs, we extracted and calculated the parameters of these rock glaciers, and analyzed the characteristics of their spatial distribution within the Daxue Shan. Our inventory suggests that the lower altitudinal boundary for permafrost across the eight aspects of slopes observed in the Daxue Shan (*i.e.*, slopes facing north, northeast, east, southeast, south, southwest, west and northwest) differs significantly. The lower altitudinal permafrost boundary is ~104 m lower on eastern- rather than western-facing slopes. The analysis of rock glaciers parameters indicates that the formation of rock glaciers is closely related to local topographical parameters. These results show that environmental controls (*i.e.*, topographical, climatic, lithological factors) greatly affect the formation and development of rock glaciers. This study provides important data for exploring the relation between maritime periglacial environments and the development of rock glaciers on the southeastern Tibetan Plateau (TP). It may also highlight the characteristics typical of rock glaciers found in a maritime setting.

**Keywords:** rock glaciers; inventory; distribution; environmental controls; Daxue Shan

**1 Introduction**

The term 'rock glacier' was first proposed by the American scholar Capps when the investigating Kennicott Glacier in Alaska (Capps, 1910). By definition, rock glaciers consist of perennially frozen masses of ice and debris that creep downslope under the weight of gravity (Haeberli, 1985; Barsch, 1996; Haeberli et al., 2006). As the bodies of rock glaciers are similar to

5 moraines in that, as their ice mass moves over a pore ice surface, they do not sort materials in relation to the thickness of the debris they contain. Many Himalayan rock glaciers develop out of moraines and it is hard to distinguish where the moraine ends and the rock glacier begins. Statistically, rock glaciers occupy extensive areas above the forest line in the mountainous regions of the world (Haeberli, 1985). Indeed, there are ~73,000 rock glaciers in the world (Jones et al., 2018a), with ~1,000 active rock glaciers in the Swiss Alps alone. The ways in which rock glaciers move can significantly influence any engineering

10 and transportation infrastructure in regions affected by permafrost. The freeze-thaw process experienced by the ice masses within rock glaciers can exert a major impact on the hydrological cycle (Azócar and Brenning, 2010; Jones et al., 2018a; Jones et al., 2018b), and is vital to understand when reconstructing the local paleoclimate and paleoenvironment. Rock glaciers are therefore not only characterized by an advanced form of creep movement, but are also complex landforms which incorporate many of the phenomena observed in ice margins. 
[revised manuscript text omitted]

10   4,300~4550 m asl belt. Tongue-shaped and lobate-shaped rock glaciers are mainly concentrated in the 4,350~4,600 m asl zone (Fig. 4a). We found that the asl altitudes of moraine-type rock glaciers were at least 50~100 m higher than for talus-derived rock glaciers, and that the lower boundaries of tongue-shaped rock glaciers were ~50 m lower than for lobate rock glaciers. The upper boundaries for the vast majority of rock glacier types were ~4,600 m asl, at higher altitude there are often present some ice glaciers. The finding that tongue-shaped rock glaciers flow further downvalley than lobate rock glaciers was also

15   verified by a comparative analysis between moraine-type, tongue-shaped rock glaciers (MTRG) versus moraine-type, lobate rock glaciers (MLRG), and talus-derived, tongue-shaped rock glaciers (TTRG) versus talus-derived, lobate rock glaciers (TLRG); the lower altitudinal boundary for MTRG and TTRG was ~100 m lower than for MLRG and TLRG. Figure 4b shows the range in areas covered by different types of rock glaciers. Apart from a few outliers, it can be seen that the area of most rock glacier types area is <0.3 km², and that, in this regard, there is no clear difference between these different rock glacier

20   types. Figure 4c shows the range in the mean gradients of the slopes of different types of rock glaciers. Moraine-type and talus-derived rock glaciers exhibit mean gradients which are all concentrated within the 22°~35° range. However, tongue-shaped and lobate rock glaciers display a greater difference in mean gradient. Tongue-shaped rock glaciers have slopes with mean gradients which are concentrated in the 22°~35° range, whereas the mean gradients of lobate rock glaciers fall within the 27°~45° range, meaning that the upper (~10°) and lower (~5°) slopes of tongue-shaped rock glaciers are both ~5° lower than

25   for lobate rock glaciers. Figure 4d displays the range in the lengths of different types of rock glaciers. Moraine-type, talustype and tongue-shaped rock glaciers are mostly 500~1000 m long, whereas  lobate rock glaciers are mostly 200~400 m long, compared with  lobate rock glaciers, moraine-type and tongue-shaped rock glaciers have more sediment supplies and last longer on gentle slope, indicating that moraine-type and tongue-shaped rock glaciers flow further than  lobate rock glaciers.

5   Our dataset revealed that, apart from south-facing (5.44%), southeast-facing (3.06%) and northeast-facing (20.75%) slopes, the rock glaciers of the Daxue Shan are fairly evenly distributed on slopes with the remaining five aspects, which each aspect accounting for ~15% of the total. Moraine-type  rock glaciers are most often northeast-facing (30.34%) and north-facing (20%), but talus-derived  rock glaciers are most often southwest-facing (22.82%) and west-facing (17.45%); they are less commonly southeast-facing (5.37%), south-facing (6.71%) and north-facing (8.72%).

10   Lobate rock glaciers tend to be found less on south-facing (6.25%) and southeast-facing (0%) slopes, but more commonly on north-facing, northwest-facing and east-facing, which each aspect accounting for ~18.75% of the total. We compared all our results and discovered that shady (*i.e.*, N, NE and E) slopes appear more conducive to the formation of moraine-type rock glaciers, and sunny (*i.e.*, W, SW and S) slopes appear more conducive to the formation of talus-derived rock glaciers. In addition, there are more steep rock walls on the north faces producing debris, north-facing (*i.e.*, N, NW and NE) slopes seem

15   to be more favorable for the formation of lobate rock glaciers than do south-facing (*i.e.*, SW, S and SE) ones (Fig. 5).

The mean altitude of a rock glacier's front (MAF) has often been taken to be a good approximation of the lower boundary of the discontinuous permafrost zone (*i.e.*, Scotti et al., 2013). We found a significant altitudinal difference between the lower permafrost boundaries identified on the abovementioned eight aspects as they were categorized for the Daxue Shan. For

20   example, permafrost was assumed to be probable above 4,300 m asl on east-facing slopes, and above 4,403 m asl on west-facing slopes. The mean lower permafrost boundary was calculated as occurring at 4,352 m asl (derived from a mean value of 4,315 m asl for east-facing slopes at 4315m, and 4,419 m asl for west-facing slopes). The mean lower permafrost boundary on east-facing (shady) slopes would therefore probable be 104 m lower than that of west-facing (sunny) slopes (Fig. 6).

**5 Discussion**

The spatial distribution and dynamics of rock glaciers are especially dependent upon the local topography and climate (Springman et al., 2012; Delaloye et al., 2013). Analyzing local environmental factors is therefore crucial to obtaining an understanding of the formation, development and spatial distribution of rock glaciers.

**5.1 Topographical controls on rock glaciers**

The results showed that there is a significantly positive correlation ($p$=0.01) between rock glacier area, length and width. We also found that latitude has a significantly positive correlation ($p$=0.01) with rock glacier length, width and area (Table 2), indicating that latitude may affect the existence of rock glaciers in the Daxue Shan. The higher the latitude becomes, the greater are the length, width and area of rock glaciers, and the more conducive is the environment to their formation and development. The spatial distribution of the rock glaciers of the Daxue Shan is therefore related to latitude. In addition, altitude asl has a significantly positive correlation ($p$=0.01) between rock glacier width and area; larger-scale rock glaciers occur mainly in the higher mountains. We also found 
[revised manuscript text omitted]

---

## Author Response (AR2)

**Reply to comments by editor on "Rock glaciers in Daxue Shan, southeastern Tibetan Plateau: an inventory, their distribution, and their environmental controls"**

We thank the editor Peter Morse for the very constructive comments and suggestions. These comments are all valuable and very helpful for revising and improving our paper, as well as the important guiding significance to our researches. We have studied comments carefully and have made corrections which we hope meet with approval. Our replies are highlighted in blue. The responds to the editor's comments are shown below.

All the best, Zeze Ran and Gengnian Liu

P1L9: Daxue Shan as a place does not need the determiner "the", but it is OK to say "the Daxue Shan region". Please check the rest of the paper, including figure or table captions.

Reply: Thank you for pointing out this and we have revised as suggested.

P1L15: Repetitive. topographical parameters are invoked twice. Re-write for clarity. Reviewer 3 asked for clarification of topographic controls. Be explicit, i.e. slope, aspect, etc.

Reply: Thank you for pointing out this and we have revised as suggested.

P3L7: Can you indicate this on one of the plates in Figure 1? Some readers may not be familiar with this area. P3L20: These areas are important, and Mt. zheduo is mentioned several times. These locations must be shown on Figure 1(c). P3L23: show on Figure 1(c)

Reply: Thank you for pointing out this. Considering that the *Yangtze Platform, Minjiang, Yarlung and Dadu rivers* only appeared once in the text, these locations were not further discussed. And also, from the research scale of this paper, the distance between the above locations and Daxue shan was relatively far beyond the map range of Figure 1(a) (b) (c). Therefore, we removed the descriptive statement (*Yangtze Platform, Minjiang, Yarlung and Dadu rivers*) that appeared only once in the text.

P4L1: Reference?

Reply: Thank you for pointing out this and we have added reference.

P5L15: Cite the map authorship here as with any publication, and add the publiation to the references.

Reply: Thank you for pointing out this and we have revised as suggested.

P7L5: This paragraph explains to the reader why you did not perform a functional analysis. It would be best to move this paragraph to the beginning of section 3.2.

Reply: Thank you for pointing out this and we have revised as suggested.

P7L11: As one reviewer pointed out, the results are often too descriptive. many of the details can be gotten from

the tables and figures. Also, the specific tables of figures can be cited more often to make specific points.

Reply: Thank you for pointing out this and we have reduced some descriptive sentences.

P7L17: P-value? what was the test?

Reply: Thank you for pointing out this. Revised data shows that there is not enough evidence and we have removed this section.

P7L20: This is discussion and should be moved to Section 5.1

Reply: Thank you for pointing out this. Revised data shows that there is not enough evidence and we have removed this section.

P9L5: This reads more like discussion and should probably be moved.

Reply: Thank you for pointing out this. This is a summary of the aspect result, which is still the summary result from the front of the aspect. Therefore, in logical relationship, we hope it can follow behind the aspect result.

P9L10: Arguably, this also reads like discussion. Present results in Results, and save the interpretation for Discussion.

Reply: Thank you for pointing out this and we have revised as suggested. (P10L11-P10L17)

P9L15: Repetitive. You lump topography and climate into environmental factors, along with lithological. These 2 sentences can be combined and written with more clarity.

Reply: Thank you for pointing out this and we have rewritten this section.

P9L18: Why is there an even distribution in aspect? You need a discussion and summary statement about aspect in the discussion. Move P7L18-P8L2 to here.

Reply: Thank you for pointing out this. Revised data shows that there is not enough evidence and we have removed this section.

P27L2: Project (a) to a suitable projection as R3 suggests and indicate the projection in the caption. Because of PFI color scale, the study area polygon is not obvious. Show study area as a black polygon. On scales, it should be km not Km. Fig. 1c should be a satellite image or an actual map. Show geographic features mentioned in the text as per comments above.

Reply: Thank you for pointing out this and we have revised as suggested.

P28L7: Check Google Earth FAQ for proper citation. You are required to acknowledge the type of images shown in the maps.

Reply: Thank you for pointing out this. We have revised as suggested and acknowledged the type of images shown in the maps once again.

P29L1: Show contours on Fig. 3 so that topographic constraint on rock glaciers is better shown in the paper. Even better, the existing Fig. 1c could become Fig. 3a if the glacier polygons are overlain in black. Alternatively, contours could also be shown on Fig. 8.

Reply: Thank you for pointing out this and we have revised as suggested.

P30L4: Use much more contrast between the fill and the line elements in the box plots. As is, the mean and median values are impossible to see. Remove "(a)", "(b)", "(c)", and "(d)" from axis titles and place horizontally within each panel. Add spaces between titles and units. E.g., "Altitude(m asl)" becomes "Altitude (m asl)". Capitalize first word of legend entries.

Reply: Thank you for pointing out this and we have revised as suggested.

P31L3: Add spaces between lables and units,e.g., "TTRG(%)" becomes "TTRG (%)". This figure cannot be published as is. It is too poor in quality. There are obvious compression artifacts and it is blurry.

Reply: Thank you for pointing out this and we have revised as suggested.

P33L3: There are numerous spaces missing from figure titles and labels that must be corrected.

Reply: Thank you for pointing out this and we have revised as suggested.

P34L1: Include lithological codes in the legend. The map must be cited here and added to the list of references.

Reply: Thank you for pointing out this and we have revised as suggested.

**The specific details are as follows with changes highlighted in yellow.**

[revised manuscript text omitted]
_3xd$ | Late Triassic slate and phyllite | $T_3bd$ | Late Triassic sandstone and conglomerate |
| $T_1b$ | Early Triassic sandstone and siltstone | $Z_1m$-sh | Lower Sinian meta-sandstone and phyllite |
| $P_2d$ | Middle Permian volcanic lava and breccia | $C_2$-$P_1xg$-s | Middle Carboniferous-Early Permian Carbonatite |
| $R\eta\gamma$ | Tertiary monzonitic granite | $P\Sigma$ | Permian unclassified ultrabasicrock |
| $T_3zh$ | Late Triassic sandstone | $Pt_1K.$ | Palaeoproterozoic amphibolite and migmatitic gneiss |
| $T\eta\gamma$ | Triassic monzonitic granite | $P_2k$ | Middle Permian carbonate and pelite |
| $T_3lh$ | Late Triassic meta-sandstone | $T_2l$ | Middle Triassic dolomite and limestone |

**Figure 8: The rock glaciers of  Daxue Shan superimposed on the local lithologic-geologic environment. Stratigraphic data from the China Geological Survey (http://www.cgs.gov.cn/).**

---

## Author Response (AR3)

**Reply to comments by editor on "Rock glaciers in Daxue Shan, southeastern Tibetan Plateau: an inventory, their distribution, and their environmental controls"**

We thank the editor Peter Morse for the very constructive comments and suggestions. These comments are all valuable and very helpful for revising and improving our paper, as well as the important guiding significance to our researches. We have studied comments carefully and have made corrections which we hope meet with approval. Our replies are highlighted in blue. The responds to the editor's comments are shown below.

All the best, Zeze Ran and Gengnian Liu

P5L12: Reference the source here. This geological data has to be something that the reader can locate, and should be treated like any publication. Referencing the website for the Chinese Geological Survey is inadequate. Refer to the source map.

Reply: Thank you for pointing out this and we have revised as suggested.

P8L12: At the end of the results, you need a line or 2 about lithology results. Fig. 6 will then have to change position and subsequent Figs. renumbered. Please then check the subsequent figure references in the text.

Reply: Thank you for pointing out this and we have revised as suggested.

P8L20: You don't give the p-values in your table. This is the α level of the test. Carefully change p-value to α-level in the text and in Table 3.

Reply: Thank you for pointing out this and we have revised as suggested.

P9L4: (Ed. note: This would be the new Fig. 3a)

Reply: Thank you for pointing out this and we have revised as suggested.

P11L12: This should stem from a result, but currently the lithological relations are not mentioned in the results section.

Reply: Thank you for pointing out this and we have revised as suggested.

P26L1: As suggested before. It would be best to show a satellite image here or a topographic map. The detailed elevation data shown here would be best shown as Figure 3a. The current Figure 3 would become Figure 3b. If you use a Google Earth image, please use the proper attribution such as: "Map data: Google, CNES/Airbus". Whatever is shown in Google Earth.

Reply: Thank you for pointing out this and we have revised as suggested.

P27L1: Please write this out as the attributions within the images are too small to read.

Reply: Thank you for pointing out this and we have revised as suggested.

P28L1: This ends up looking too messy, please remove the contours. Please move your current Fig 1c here to become Fig. 3a, and this figure becomes fig 3b. Overlay the RG on both panels, and the reader will very clearly see the relations between elevation and PZI. The caption will have to be modified accordingly, as will figure references in the text.

Reply: Thank you for pointing out this and we have revised as suggested.

P29L1: Thank you for increasing the contrast of the line work. I further suggest separating the blue classes into different colors. In particular, "Moraine-type-and Tongue-shaped RG" looks like "Talus-type and tongue-shaped RG".

The legend is also blurry and shows jpeg compression artifacts. Please clear this up.

Reply: Thank you for pointing out this and we have revised as suggested.

P31L1: Pleas remove the rectangular outlines around the group labels. These outlines just add clutter to the figure.

Reply: Thank you for pointing out this and we have revised as suggested.

P32L1: Please remove the labels from the Temperature series. They are unnecessary.

Reply: Thank you for pointing out this and we have revised as suggested.

P33L1: This legend is still not ready. The key table and legend entries do not match. As requested previously, please edit the legend so that the stratigraphic codes are shown, and delete the table. This is too general. The reader needs to be able to get a hold of the map data, and this is inadequate. If the map has authorship, cite here as for any publication, and add the reference to your list.

Reply: Thank you for pointing out this and we have revised as suggested.

**The specific details are as follows with changes highlighted in yellow.**

[revised manuscript text omitted]

Legend:
- Moraine-type RG (146)
- Talus-type RG (149)
- Tongue-shaped RG (279)
- Lobate-shaped RG (16)
- Moraine-type and tongue-shaped RG (139)
- Moraine-type and lobate-shaped RG (7)
- Talus-type and tongue-shaped RG (140)
- Talus-type and lobate-shaped RG (9)

[Figure]

**Figure 5: Analysis of the abundances of different rock glacier types versus aspect. The number of rock glaciers for each aspect on each of the four radar plots is shown as a percentage (%). (Note: RG=rock glaciers; MTRG= moraine-type and tongue-shaped rock glaciers; MLRG= moraine-type and lobate rock glaciers; TTRG=talus-derived and tongue-shaped rock glaciers; TLRG= talus-derived and lobate rock glaciers)**

[Figure]

**Figure 6: The rock glaciers of Daxue Shan superimposed on the local lithologic-geologic environment. Stratigraphic data from the Li et al. (1999).**

[Figure]

**Figure 7:** Minimum altitudinal rock glacier fronts (MAF) for all eight aspects, along with the overall mean. These values are taken to represent the lower boundaries of the potential permafrost extent in the Daxueshan region (bars indicate standard errors of the mean). Because Daxue Shan lies along an approximately NW-SE axis, we used this NW-SE axis as the boundary separating east-facing (*i.e.*, N, NE, E), shady slopes from west-facing (*i.e.*, S, SW, W), sunny slopes.

[Figure]

**Figure 8: Climatographs for the Kangding (2,615.7 m asl, 30.03°N, 101.58°E), Daofu (2,957.2 m asl, 30.59°N, 101.07°E), Danba (1,949.7 m asl, 30.53°N, 101.53°E) and Ganzi (3,393.5 m asl, 31.37°N, 100°E) meteorological stations. Data sources: Meteorological Data Center of the China Meteorological Administration (http://data.cma.cn/, calculated for the period 1981–2010, inclusive).**

---

## Editor Decision (ED3)

[revised manuscript text omitted]
  | 1.000    | -0.893** | 0.116*   | 0.102*        | -0.020  | 0.029    | 0.092            | - <del>0.016</del> |
| Longitude | -0.893** | 1.000           | -0.290** | -0.062 | 0.025   | 0.002    | -0.004           | <del>-0.03</del> 4 |
| Altitude  | 0.116*   | -0.290**        | 1.000    | -0.075 | 0.087   | 0.031    | -0.102*          | <del>0.045</del>   |
| Length    | 0.102*   | -0.062          | -0.075   | 1.000         | 0.063   | 0.776**  | -0.341**         | 0.013              |
| Width     | -0.020   | 0.025           | 0.087    | 0.063         | 1.000   | 0.572**  | -0.004           | - <del>0.026</del> |
| RG area   | 0.029    | 0.002           | 0.031    | 0.776**       | 0.572** | 1.000    | - 0.265** | 0.010              |
| slope     | 0.092    | -0.004          | -0.102*  | -0.341**      | -0.004  | -0.265** | 1.000            | -0.068             |
| Aspect    | -0.016   | -0.034          | 0.045    | 0.013         | -0.026  | 0.010    | -0.068           | 1.000              |

---

## Author Response (AR4)

**Reply to comments by editor on "Rock glaciers in Daxue Shan, southeastern Tibetan Plateau: an inventory, their distribution, and their environmental controls"**

We thank the editor Peter Morse for the very constructive comments and suggestions. These comments are all valuable and very helpful for revising and improving our paper, as well as the important guiding significance to our researches. We have studied comments carefully and have made corrections which we hope meet with approval. Our replies are highlighted in blue. The responds to the editor's comments are shown below.

All the best, Zeze Ran and Gengnian Liu

P26L1: The image resolution is very low. When I look at the area in Google Earth the image is much less pixelated (even the north arrow is pixelated in this figure). Increase the resolution to 600-800 dpi.

Tic marks are missing next to the lat./long. labels.

Remove the town and mountain labels from Figures 3a and 3b, and add them here instead. Use either black or white letters, whichever stands out the most for the given background. White letters will stand out against the lowlands, but not the mountain tops, whereas the reverse is true for black letters.

Reply: Thank you for pointing out this and we have revised as suggested.

P31L1: Change faults to black lines so that they are differentiated from simple lithological unit boundaries.

Reply: Thank you for pointing out this and we have revised as suggested.

P33L1: Change these from "a." to "(a)", etc., so that these match the style of other figures.

Reply: Thank you for pointing out this and we have revised as suggested.

**The specific details are as follows with changes highlighted in yellow.**

[revised manuscript text omitted]